



# SNICAR-AD v3: A Community Tool for Modeling Spectral Snow Albedo

Mark G. Flanner[1], Julian Arnheim[2], Joseph M. Cook[3], Cheng Dang[4], Cenlin He[5], Xianglei Huang[1], Deepak Singh[6], S. McKenzie Skiles[7], Chloe A. Whicker[1], and Charles S. Zender[8]

[1]Department of Climate and Space Sciences and Engineering, University of Michigan, Ann Arbor, MI, USA
[2]Department of Earth and Atmospheric Sciences, Cornell University, Ithaca, NY, USA
[3]Department of Environmental Science, Aarhus University, Roskilde, Denmark
[4]Joint Center For Satellite Data Assimilation, UCAR, Boulder, CO, USA
[5]Research Applications Laboratory, National Center for Atmospheric Research, Boulder, CO, USA
[6]Centre of Studies in Resources Engineering, Indian Institute of Technology Bombay, Mumbai, Maharashtra, India
[7]Department of Geography, University of Utah, Salt Lake City, UT, USA
[8]Department of Earth System Science, University of California, Irvine, CA, USA

**Correspondence:** Mark Flanner (flanner@umich.edu)

**Abstract.** The Snow, Ice, and Aerosol Radiative (SNICAR) model has been used in various capacities over the last 15 years to model the spectral albedo of snow with light-absorbing constituents (LAC). Recent studies have extended the model to include an adding-doubling two-stream solver and representations of non-spherical ice particles, carbon dioxide snow, snow algae, and new types of mineral dust, volcanic ash, and brown carbon. New options also exist for ice refractive indices and solar zenith

angle-dependent surface spectral irradiances used to derive broadband albedo. The model spectral range was also extended deeper into the ultraviolet for studies of extraterrestrial and high-altitude cryospheric surfaces. Until now, however, these improvements and capabilities have not been merged into a unified code base. Here, we document the formulation and evaluation of the publicly-available SNICAR-ADv3 source code, web-based model, and accompanying library of particle optical properties. The use of non-spherical ice grains, which scatter less strongly into the forward direction, reduce the simulated albedo

perturbations from LAC by $\sim 9 - 31\%$, depending on which of the three available non-spherical shapes are applied. The model compares very well against measurements of snow albedo from seven studies, though key properties affecting snow albedo are not fully constrained with measurements, including ice effective grain size of the top sub-millimeter of the snowpack, mixing state of LAC with respect to ice grains, and site-specific LAC optical properties. The new default ice refractive indices produce extremely high pure snow albedo ($> 0.99$) in the blue and ultraviolet part of the spectrum, with such values measured so far

only in Antarctica. More work is needed particularly in the representation of snow algae, including experimental verification of how different pigment expressions and algal cell concentrations affect snow albedo. Representations and measurements of the influence of liquid water on spectral snow albedo are also needed.



## 1 Introduction

Snow is among the most reflective natural surfaces on Earth and therefore plays an important role in determining its climate state. The albedo of snow is determined by many factors, including the morphology and size of the ice grains, the spectral and directional distribution of incident sunlight, and the content of light-absorbing constituents (LAC). Spectral snow albedo simulations are applied to represent snow albedo feedback and cryospheric influence on the planetary energy budget, quantify impacts of anthropogenic pollution and other natural substances on snow albedo and radiative forcing, and educate people

on snow physics and radiative transfer. The purpose of this document is to describe the technical formulation, accompanying library of particle optical properties, and evaluation of a spectral snow albedo model — The Snow, Ice, and Aerosol Radiative model with Adding-Doubling solver, version 3.0 (SNICAR-ADv3). A single layer version of this model can be run online at: http://snow.engin.umich.edu and accompanying source code for the multi-layer column model is linked to under *Code and data availability*.

A variety of techniques that account for multiple scattering by ice grains have been employed to simulate snow albedo. Wiscombe and Warren (1980) and Warren and Wiscombe (1980) combined a two-stream radiative transfer solution with the delta-Eddington approximation and Mie solutions to simulate hemispheric albedo of a single snow layer of any thickness and grain size, overlying a surface with any albedo, and including the influence of light-absorbing particles. Directional and hemispheric reflectance of snow has been simulated with multi-stream discrete ordinate approximations (Grenfell et al., 1994;

Nolin and Dozier, 2000; Painter and Dozier, 2004; Gardner and Sharp, 2010; Dang et al., 2019). Directional snow reflectance has also been simulated with Monte-Carlo photon tracking techniques (Kaempfer et al., 2007; Picard et al., 2009; Dumont et al., 2010; Schneider et al., 2019; Larue et al., 2020), adding-doubling solutions (Sergent et al., 1998; Leroux et al., 1998; Aoki et al., 1999, 2000; Li and Zhou, 2004), and analytic solutions of the radiative transfer equation under idealized conditions (Mishchenko et al., 1999; Dumont et al., 2010). Libois et al. (2013) developed a multi-layer two-stream snow albedo model

utilizing formulations from Kokhanovsky and Zege (2004) that explicitly account for non-spherical ice particles, thereby improving simulation of the vertical profile of light extinction in snow.

The original SNICAR code combined theory from Wiscombe and Warren (1980) and Warren and Wiscombe (1980) with the multi-layer two-stream solution from Toon et al. (1989), and was introduced by Flanner and Zender (2005) to improve the simulation of snow albedo and the vertical distribution of solar absorption in the NCAR Community Climate System Model.

Subsequent studies have explored radiative and climate impacts of snow-deposited light-absorbing particles using broadband implementations of SNICAR in global and regional climate models (e.g., Flanner et al., 2007, 2009; Bond et al., 2011; Qian et al., 2011; Lawrence et al., 2012; Flanner et al., 2012; Jiao et al., 2014; Zhao et al., 2014; Lin et al., 2014; Oaida et al., 2015; Wu et al., 2018; Singh et al., 2018; Matsui et al., 2018; Ward et al., 2018; Li and Flanner, 2018), and using high spectral resolution single-column SNICAR calculations (e.g., McConnell et al., 2007; Ming et al., 2008; Yasunari et al., 2011; Kaspari

et al., 2011; Hadley and Kirchstetter, 2012; Painter et al., 2013; Sterle et al., 2013; Young et al., 2014; Polashenski et al., 2015; Singh and Flanner, 2016; Schmale et al., 2017; Skiles et al., 2017; Torres et al., 2018; He et al., 2018; Skiles and Painter, 2018;



Pu et al., 2019; Gleason et al., 2019; Skiles and Painter, 2019; Kaspari et al., 2020; Uecker et al., 2020; Gelman Constantin et al., 2020).

In 2011, a web-based 470-band implementation of SNICAR was launched for informal educational purposes. This web model was executed more than 185,000 times between June 2011 and June 2020 by users from dozens of countries and became more widely used than anticipated. During this period there have been numerous improvements to the model (e.g., Flanner et al., 2014; Singh and Flanner, 2016; Cook et al., 2017; He et al., 2018; Skiles et al., 2017; Dang et al., 2019), but they have not been assimilated into a unified code base. Furthermore, model users have requested improved documentation of the web-based model and accompanying code. Together, this motivated the creation and release of SNICARv3 in June 2020 and SNICAR-ADv3 in January 2021, along with this study.

Dang et al. (2019) compared several two-stream models of snow albedo, including SNICAR, with 16-stream solutions from the Discrete Ordinates Radiative Transfer (DISORT) model. They found that the delta-Eddington adding-doubling two-stream approximation from Briegleb (1992) and Briegleb and Light (2007) produces the most accurate albedo, especially in the near-infrared (NIR) spectrum under diffuse light (cloudy) conditions. The adding-doubling scheme also allows for incorporation of internal Fresnel layers, enabling representation of ice and ponded ice surfaces (e.g., Briegleb and Light, 2007). For these reasons, Dang et al. (2019) proposed merging features of SNICAR with the adding-doubling solver, branded as "SNICAR-AD", and they have since incorporated a broadband version of this model into the Exascale Earth System Model (E3SM). Here, we describe numerous additions and improvements to the single-column, high-spectral resolution version of SNICAR, including new options for the representation of non-spherical ice particles (He et al., 2017, 2018), snow algae (Cook et al., 2017), carbon dioxide ice (Singh and Flanner, 2016), $H_2O$ ice refractive indices (Picard et al., 2016), dust optical properties suitable for Earth and Mars (Skiles et al., 2017; Polashenski et al., 2015; Wolff et al., 2009), brown carbon optical properties (Kirchstetter et al., 2004), larger dust and volcanic ash particles, and surface spectral irradiance profiles for different atmospheric conditions that depend on solar zenith angle (SZA). We have also extended the spectral range of the model into the ultraviolet (UV) spectrum to a wavelength of $0.2\,\mu m$.

This paper focuses on the simulation of dry snow albedo by SNICAR-ADv3. A companion study (*Whicker et al.*, in prep.) describes and evaluates an extension of this model for the simulation of glacier ice. This paper also focuses on narrowband (10 nm resolution) simulations. Approaches for adapting narrowband models like SNICAR to simulate broadband fluxes in ESMs are explored elsewhere (e.g., Lawrence et al., 2018; Aoki et al., 2011; Dang et al., 2019; van Dalum et al., 2019). Finally, we also raise awareness of other publicly-available snow albedo models. The Two-streAm Radiative TransfEr in Snow (TARTES) model (Libois et al., 2013, 2014) can be run online at: http://snowtartes.pythonanywhere.com/. A web-based model that provides the albedo of sloped snow surfaces (Picard et al., 2020), dependent on illumination and slope geometry, is at: http://snowslope.pythonanywhere.com/.

This paper is organized as follows: Sect. 2 describes the bulk model formulation, including surface spectral irradiance options for the upper model boundary. Sect. 3 describes the constituent optical properties, including ice, light-absorbing particles, and snow algae. Sect. 4 describes modeled spectral and broadband albedo sensitivities to different environmental features, model





options, and LAC concentrations. Finally, Sect. 5 presents evaluations of the model against field measurements of spectral snow albedo from different environments.

## 2 Model Formulation

### 2.1 Two-stream solution

SNICAR and other two-stream models of solar radiation require inputs of SZA, downwelling spectral irradiance at the column upper boundary, and albedo of the underlying substrate at the column lower boundary. Additionally, the following spectrally-dependent bulk properties must be defined for each model layer:

- Extinction optical thickness ($\tau$). Range: 0 to $\infty$

- Single-scatter albedo ($\omega$): the probability that a photon experiencing an extinction event is scattered as opposed to
absorbed. Range: 0 to 1

- Scattering asymmetry parameter ($g$): the average cosine of the scattering phase angle. Range: $-1$ to 1

Multi-layer two-stream models utilize these fundamental quantities to solve for the upward and downward radiative fluxes at each layer interface within each spectral band, from which spectral albedo at the model top, radiative absorption within each layer, and spectral transmittance through the column are derived. Ice grains scatter strongly in the forward direction because they are generally much larger than the wavelengths of interacting light. Strong forward peaks in the scattering phase functions
necessitate transformations or analytic manipulations of the two-stream input variables to accurately represent fluxes (e.g., Joseph et al., 1976; Wiscombe and Warren, 1980; Bohren and Huffman, 1983). The delta scalings of $\tau$, $\omega$, and $g$ (Joseph et al., 1976, Appendix A) are applied to account for forward scattering in all versions of SNICAR.

While previous versions of SNICAR adopted a tri-diagonal matrix two-stream solver (Toon et al., 1989), SNICAR-ADv3
utilizes an adding-doubling solution (Briegleb, 1992; Briegleb and Light, 2007), which has several advantages (Dang et al., 2019). First, the adding-doubling framework enables internal Fresnel layers to be incorporated into the model, paving the way for unified treatment of snow and ice. Second, compared with 16-stream albedo solutions, the delta-Eddington adding-doubling approximation provides more accurate albedo estimates, especially under diffuse conditions (Dang et al., 2019). Third, the adding-doubling solution is stable under all conditions except when SZA$= 90°$ (i.e., when the sun is exactly on the horizon),
whereas the tri-diagonal matrix formulation can encounter rare singularities across a broader range of SZA, dependent on the column optical properties and the approximation applied. Fourth, the adding-doubling solver is more computationally efficient than the formulation from Toon et al. (1989). For these reasons, we have transitioned SNICAR to the adding-doubling solver, i.e., SNICAR-AD (Dang et al., 2019). SNICAR-ADv3 combines delta scalings with the Eddington two-stream approximation (Briegleb and Light, 2007), whereas previous versions of SNICAR combined the delta scalings with either the Hemispheric-
mean or Eddington two-stream approximations as formulated by Toon et al. (1989). Another difference between these solutions is that fluxes under diffuse illumination are solved for via angular integration of direct-beam incidence at eight Gaussian points





in SNICAR-AD (e.g., Briegleb and Light, 2007), whereas diffuse incident flux is input as a distinct upper-boundary term in the solution from Toon et al. (1989). The two-stream equations applied in SNICAR-ADv3 are listed in Appendix A. Because of the mathematical singularity that occurs at $\cos(\text{SZA}) = 0$, the allowable SZA range is limited to $0 - 89°$.

## 2.2 Spectral grid

The new model simulates albedo across a wavelength ($\lambda$) range of $0.2 - 5.0\,\mu\text{m}$ at $10\,\text{nm}$ resolution. There are 480 bands in total, centered at: [$205\,\text{nm}$, $215\,\text{nm}$, $225\,\text{nm}$, ..., $4995\,\text{nm}$]. The model was extended from a lower wavelength bound of $300\,\text{nm}$ to $200\,\text{nm}$ to handle more of the UV spectrum. Though there is insufficient surface irradiance at wavelengths of $200 - 300\,\text{nm}$ to appreciably affect broadband albedo of typical Earth surfaces, surface albedo in this spectral range is important for studies of Mars' polar ice caps (Singh and Flanner, 2016; Singh et al., 2018), high-altitude regions of Earth, and photochemical reactions on Earth and other planets (e.g., Singh, 2020). All files in the accompanying library of particle optical properties for SNICAR-ADv3 are defined on this spectral grid, though many of these definitions require uncertain extrapolation to the shortest and longest wavelengths.

### 2.3 Bulk layer properties

The spectrally-dependent bulk layer properties that enter the two-stream solver depend on spectrally-resolved optical properties and mixing ratios of all constituents present within the layer. The layer extinction optical depth of constituent $n$ depends on its mass extinction cross-section ($k_{e,n}$, units of $\text{m}^2\,\text{kg}^{-1}$) and layer mass burden ($L_n$, units of $\text{kg}\,\text{m}^{-2}$) as:

$$\tau_n = k_{e,n}L_n \tag{1}$$

The layer burden of each constituent depends on its mass mixing ratio ($q_n$, units of $\text{kg}_\text{n}\,\text{kg}_\text{snow}^{-1}$) as:

$$L_n = \Delta z \rho_s q_n \tag{2}$$

where $\Delta z$ is the snow layer thickness and $\rho_s$ is the density of snow in the layer. Here, the mass of snow and $\rho_s$ include the mass of ice and all other constituents present in the snow. Note that in previous versions of SNICAR, mass mixing ratio was defined as the constituent mass per unit mass of ice. Differences between these two definitions are negligible for the trace mixing ratios of impurities ($10^{-9}$ to $10^{-4}$) that are typically specified in the model. The definition of $q$ applied here is a true mixing ratio so that $\sum q_n = 1$ and enables users to simulate the albedo of non-ice substrates by specifying $q = 1$ instead of $q = \infty$. It also conforms with a definition of snow density consistent with that typically measured in the field, e.g., using a mass measurement that includes all constituents. Ice grains are therefore treated as any other constituent in this representation, but instead of querying for a mass mixing ratio of ice, the mass burden of ice is derived from the user-specified snow density by assuming all non-impurity mass of snow is ice:

$$L_\text{ice} = \Delta z \rho_s - \sum_{n=2}^{N} L_n \tag{3}$$



where ice is identified as constituent $n = 1$ of $N$ total constituents.

With $\tau_n$ defined for each constituent, including ice, the layer extinction optical depth is the sum from that of all $N$ constituents:

$$\tau = \sum_{n=1}^{N} \tau_n \tag{4}$$

The bulk layer single-scatter albedo and scattering asymmetry parameter are calculated with optical depth weighting and scattering optical depth weighting, respectively, of each constituent:

$$\omega = \frac{\sum_{n=1}^{N} \tau_n \omega_n}{\sum_{n=1}^{N} \tau_n} \tag{5}$$

$$g = \frac{\sum_{n=1}^{N} \tau_n \omega_n g_n}{\sum_{n=1}^{N} \tau_n \omega_n} \tag{6}$$


The optical properties for ice and other constituents included with SNICAR-ADv3 are described in Sect. 3.

## 2.4   Broadband albedo and surface irradiance

The code and web-based model provide spectral albedo, $\alpha(\lambda)$, calculated in each of the 480 spectral bands, and solar broadband albedo ($\overline{\alpha}$), which is weighted by surface spectral irradiance, $F^{\downarrow}(\lambda)$, as follows:

$$\overline{\alpha} = \frac{\int_{0.2\,\mu m}^{5.0\,\mu m} \alpha(\lambda) F^{\downarrow}(\lambda) d\lambda}{\int_{0.2\,\mu m}^{5.0\,\mu m} F^{\downarrow}(\lambda) d\lambda} \tag{7}$$


As a practical matter, surface irradiance band fractions are applied and provided in the web-based model output. These fractions are normalized to sum to 1.0 over the SNICAR spectral range such that broadband albedo is simply:

$$\overline{\alpha} = \sum_{i=1}^{480} \alpha_i f_i^{\downarrow} \tag{8}$$

where $\alpha_i$ is the albedo of band $i$ and $f_i^{\downarrow}$ is the fraction of surface irradiance within band $i$.

Several options for spectral irradiance are included in the SNICAR-ADv3 package, derived using different atmospheric profiles listed in Table 1. Surface spectral irradiances are calculated separately from SNICAR with the DISORT-based Shortwave Narrowband (SWNB2) model (Zender et al., 1997; Zender, 1999), using standard atmospheric vertical profiles of water vapor, ozone, and other gases, and a lower-boundary spectral albedo typical of a snowpack with an effective grain size of $100\,\mu m$. High-altitude options are provided for Summit, Greenland and a generic "high mountain" environment, which are derived by

truncating the sub-Arctic summer and mid-latitude profiles to have surface pressures of $796\,hPa$ and $556\,hPa$, respectively. We also provide a top-of-atmosphere irradiance option, which is useful for modeling surfaces of Mars (e.g., Singh and Flanner, 2016) and other bodies in our solar system with thin atmospheres, and for exploring impacts of atmospheric attenuation on





**Table 1.** Atmospheric Profile Properties

| Environment | Surface Pressure [hPa] | Water vapor mass path [kg m$^{-2}$] | Base case[a] $\overline{\alpha}$ Clear-sky (SZA=60°) | Base case[a] $\overline{\alpha}$ Cloudy-sky |
|---|---|---|---|---|
| Mid-latitude winter | 1018 | 8.5 | 0.848 | 0.913 |
| Mid-latitude summer | 1013 | 29.1 | 0.861 | 0.918 |
| Sub-Arctic winter | 1013 | 4.2 | 0.841 | 0.910 |
| Sub-Arctic summer | 1010 | 20.8 | 0.857 | 0.917 |
| Summit Greenland | 796 | 8.4 | 0.847 | 0.912 |
| High Mountain | 556 | 1.9 | 0.832 | 0.906 |
| Top-of-Atmosphere | 0 | 0 | 0.808 | — |

[a] Base case albedo parameters are listed in Table 3. The fine-grain ($r_e = 100\,\mu m$) base case is applied here.

broadband albedo. The top-of-atmosphere spectral irradiances used to drive SWNB2 are from Matthes et al. (2017), averaged over three solar cycles. The percentages of total solar irradiance in this dataset residing outside the SNICAR spectral range, at

$\lambda < 0.2\,\mu m$ and $\lambda > 5.0\,\mu m$ are only, respectively, $0.013\%$ and $0.066\%$. Surface spectral irradiances and $\overline{\alpha}$ can be somewhat sensitive, however, to the radiative transfer model and top-of-atmosphere irradiance data used (Bair et al., 2019), and we plan to investigate this more in future work.

Spectral irradiances associated with clear-sky or cloudy atmospheres are selected internally in the model to match the user-specification of direct or diffuse incident light, respectively. The cloudy irradiances are modeled with a liquid cloud of optical

thickness 10 at $\lambda = 500\,nm$, located at a pressure of $800\,hPa$ or in the bottom-most atmospheric layer of profiles with surface pressure less than $800\,hPa$. The cloud droplet properties vary spectrally based on Mie calculations of water spheres with effective radius of $10\,\mu m$. Clear-sky irradiances are calculated for the full range of SZA ($0 - 89$ degrees at 1 degree resolution) for each profile. The aerosol optical depth is fixed at $0.05$ in all profiles. Irradiance band fractions for all atmospheric profiles and SZAs are included as a package of netCDF files in the accompanying library, and a small sample of these are shown in

Fig. 1.

## 3 Constituent optical properties

### 3.1 H$_2$O ice

Ice grain optical properties are derived from Mie calculations of ice spheres and adjustments for scattering by non-spherical particles (Fu, 2007; He et al., 2017). Users specify one of three datasets of ice refractive indices, leading to retrieval of distinct

Mie properties from the accompanying library of optical properties in netCDF format. The complex refractive indices ($M = m_r + m_i i$) from the original data are linearly interpolated to the SNICAR spectral grid, except as described below. The H$_2$O ice refractive index options in SNICAR-ADv3 are:

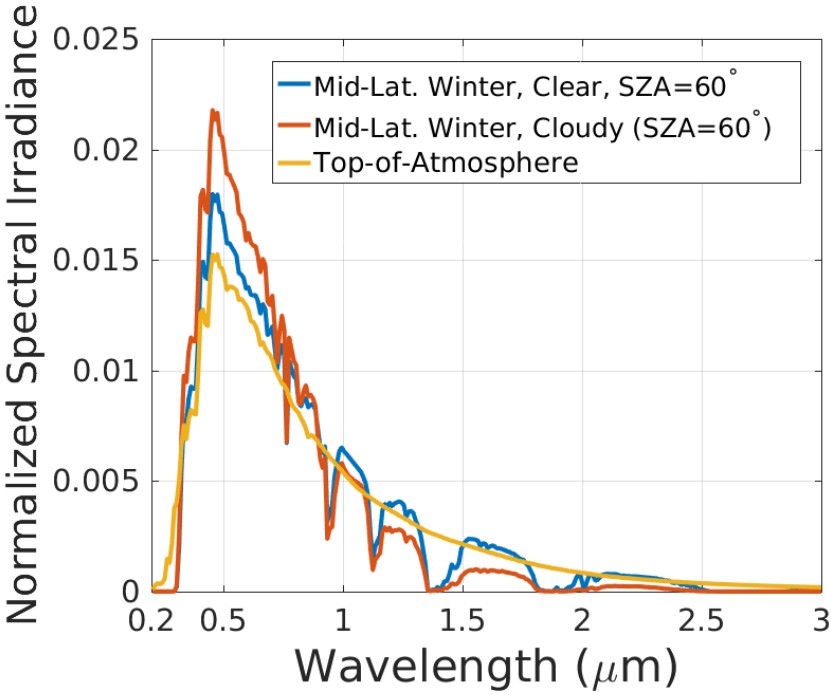

**Figure 1.** Normalized spectral irradiances for select environments. Values represent the fraction of total 0.2–5.0 μm surface irradiance within each 10 nm band, used as weights to determine broadband albedo. Top-of-atmosphere spectral irradiances at wavelengths outside those depicted constitute less than 1% of total irradiance.

1. Warren (1984) / Perovich and Govoni (1991): $m_r$ data from Warren (1984) are applied across the solar spectrum. $m_i$ data from Perovich and Govoni (1991) are applied over $\lambda = 250 - 400$ nm and data from Warren (1984) are applied at all other wavelengths. The data points from Warren (1984) at 210 and 250 nm are adjusted slightly to achieve a smooth transition in interpolated data between the two datasets over $\lambda = 210 - 250$ nm.

2. Warren and Brandt (2008): $m_r$ and $m_i$ data compiled by Warren and Brandt (2008) are applied across the solar spectrum. Note that the ice absorption data over $\lambda = 600 - 1400$ nm in this compilation originate from Grenfell and Perovich (1981).

3. Picard et al. (2016) / Warren and Brandt (2008): $m_r$ data from Warren and Brandt (2008) are applied across the solar spectrum. The 320–600 nm $m_i$ data from Picard et al. (2016) are adopted, and data from Warren and Brandt (2008) are applied at longer wavelengths. The 320 nm value reported by Picard et al. (2016) is also applied constantly over 200–320 nm, thus retaining the same feature of constancy over this spectral range that is exhibited in the Warren and Brandt (2008) dataset.





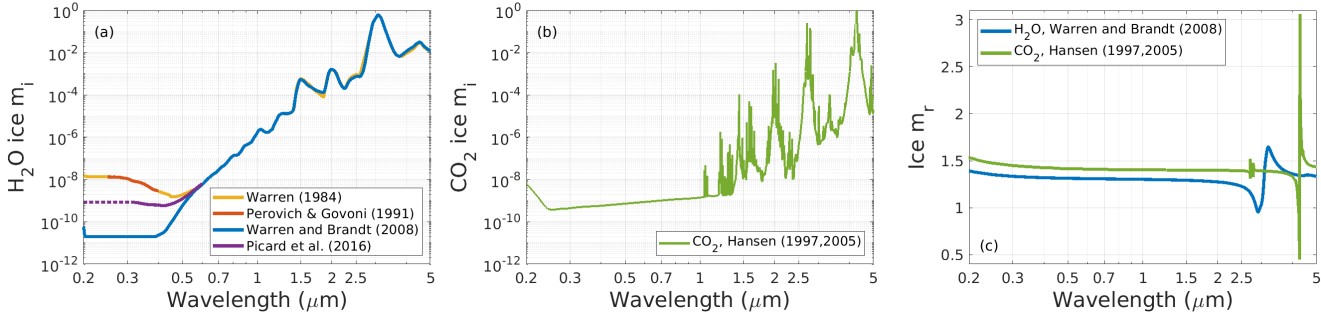

**Figure 2.** Ice complex refractive index data used in SNICAR-ADv3, including imaginary components ($m_i$) of $H_2O$ ice (a), imaginary components ($m_i$) of $CO_2$ ice (b), and real components ($m_r$) of both types of ice (c).

Values of $m_i$, which govern ice absorptivity, differ by two orders of magnitude or more in the short blue and UV part of the spectrum (Fig. 2a) across these three datasets, but are very similar in the NIR (defined here as $\lambda = 0.7 - 5.0\,\mu m$), where ice is absorptive. UV and blue ice absorptivity is extraordinarily low and notoriously challenging to measure because very long optical path lengths are needed. Dataset #1 is largely superseded by Dataset #2, the more recent and comprehensive compilation from Warren and Brandt (2008), but is retained because the original version of SNICAR applied these data, and

they arguably still represent an upper bound of ice absorptivity. Warren and Brandt (2008) argue that pure ice $m_i$ is *no larger than* $2 \times 10^{-11}$ at wavelengths of 200–390 nm, whereas Picard et al. (2016) present a central estimate and uncertainty range that includes larger (more absorptive) values than this bound. Dataset #3, the merged compilation from Picard et al. (2016) and Warren and Brandt (2008), exhibits intermediate absorptivity of the three options, and is the default option in SNICAR-ADv3. There is much less uncertainty in $m_r$, and agreement is excellent between Warren (1984) and Warren and Brandt (2008) in the

solar spectrum (Fig. 2c). (Note that these and subsequent spectral figures use log-spacing to emphasize the shorter wavelengths which contribute disproportionately to broadband albedo.)

    Mie properties are calculated using the solver from Bohren and Huffman (1983) for a large range of ice grain effective radii ($r_e$, or surface-area weighted mean radius), and using each of the three sets of $H_2O$ ice refractive indices. The optics library includes individual files for $r_e$ values ranging from 30–1500 μm, discretized by 1 μm. Properties are first calculated

for 5000 individual spheres with log-spaced radii ranging from 0.05 to 10000 μm, then weighted according to a log-normal size distribution for each of the specified $r_e$ values. The relative number of particles with radius $r$, $n(r)$, in the log-normal distribution is:

$$n(r) = \frac{1}{\sqrt{2\pi}\, r \ln \sigma_g} \exp\left[-\frac{1}{2}\left(\frac{\ln(r/r_n)}{\ln \sigma_g}\right)^2\right] \tag{9}$$

    where $\sigma_g$ is the geometric standard deviation of the size distribution, fixed at 1.5 for all of our ice particle calculations, and

$r_n$ is the number-median radius of the size distribution, related analytically to $r_e$ as:

$$r_n = r_e \exp\left[-\frac{5}{2}(\ln \sigma_g)^2\right] \tag{10}$$





The resolved effective radius of the discretized distribution is:

$$r_e = \frac{\sum r^3 n(r)}{\sum r^2 n(r)} \tag{11}$$

Comparisons of the user-specified and resolved effective radii for our calculations show near-perfect agreement, indicat-
ing that the size distributions are adequately resolved with these model parameters. We assume a bulk $H_2O$ ice density of
$917\,\mathrm{kg\,m^{-3}}$ for deriving mass-normalized optical properties.

Finally, users can specify one of four ice particle shapes:

1. Spheres

2. Spheroids (default aspect ratio: 0.5)

3. Hexagonal plates (default aspect ratio: 2.5)

4. Koch snowflakes (default aspect ratio: 2.5)

Optical properties used for spheres are simply the Mie properties of the user-specified $r_e$. For the other shapes, the user-
specified $r_e$ represents the radius of the sphere with equivalent specific surface area (SSA, or surface-area to volume ratio)
as that of the non-spherical particle (He et al., 2017). Because equal-SSA spheres provide good proxies for the extinction
cross-section and single-scatter albedo of non-spherical particles (e.g., Grenfell et al., 2005), Mie-generated $k_e$ and $\omega$ values
are also retrieved for the non-spherical particles. The scattering asymmetry parameter ($g$), however, is adjusted using the
parameterization described by He et al. (2017) and He et al. (2018), structured on the scheme developed by Fu (2007).

Corrections to $g$ are parameterized as a function of wavelength, user-specified effective radius, particle shape, and particle
aspect ratio (the ratio of grain width to length). The asymmetry parameter for hexagonal plates ($g_{\mathrm{hex}}$) follows Fu (2007):

$$g_{\mathrm{hex}} = \frac{1 - g'}{2\omega} + g' \tag{12}$$

where $g'$ is parameterized as a function of aspect ratio using the wavelength-dependent coefficients listed in Tables 1 and 2 of
Fu (2007). For other shapes, $g$ is derived from $g_{\mathrm{hex}}$ with:

$$g = g_{\mathrm{hex}} C \tag{13}$$

where $C$ is a correction factor that depends on $r_s$, the sphere radius with the same orientation-averaged projected area-to-
volume ratio of the non-spherical particle, and on the particle shape factor ($SF$), defined as the ratio of $r_s$ of the nonspherical
grain to that of an equal-volume sphere:

$$C = a_0 \left( \frac{SF}{SF_{\mathrm{hex}}} \right)^{a_1} (2r_s)^{a_2} \tag{14}$$

where $a_{0-2}$ are wavelength-dependent coefficients listed in Table 3 of He et al. (2017). Note that for convex shapes (here,
spheroids and hexagonal plates) $r_s = r_e$ and for concave shapes $r_s > r_e$ in general, and for Koch snowflakes in particular
$r_s = r_e/0.544$ (He et al., 2018).





These empirical parameterizations are based on detailed geometric optics ray-tracing calculations (Fu, 2007) that include surface wave interactions and diffraction (Liou et al., 2011, 2014; Liou and Yang, 2016). The parameterized values of $g$ at 8 wavelengths are then interpolated to the SNICAR spectral grid using a shape-preserving piecewise interpolation. Within the code, users can alter the aspect ratio (within the range of $0.1-20$) and shape factor, but the web-based model adopts the default

aspect ratios listed above and associated shape factors, which are presented in Table 1 of He et al. (2017). The parameterized value of $g$ can exceed 1.0 in rare instances with large spheroids, and consequently $g$ is capped at 0.99. Implications of particle shape for simulated albedo are described in Sect. 4.1.

### 3.2  CO$_2$ ice

To facilitate the simulation of carbon dioxide ice surfaces like those found on Mars, we also include optical properties for

CO$_2$ ice grains, as applied in SNICAR by Singh and Flanner (2016) and Singh et al. (2018). We apply concatenated CO$_2$ ice refractive indices over wavelengths of 0.2–1.8 μm and 1.8–5.0 μm from Hansen (2005) and Hansen (1997), respectively. Fig. 2b depicts the $m_i$ values from this dataset. We then apply the Kramers-Kronig relationship to derive $m_r$ values across the solar spectrum, depicted in Fig. 2c along with the H$_2$O $m_r$ values. As with H$_2$O ice, CO$_2$ ice absorbs very weakly in the visible and UV spectrum and $m_i$ values are somewhat uncertain. CO$_2$ ice $m_i$ is relatively featureless at wavelengths shorter than 1 μm

and reaches a minimum near 250 nm (Hansen, 2005). Unlike H$_2$O, however, CO$_2$ ice $m_i$ is punctuated by numerous sharp absorption bands throughout the NIR, though averaged over the NIR CO$_2$ ice is less absorptive than H$_2$O ice.

Optics files are included in the library for CO$_2$ $r_e$ ranging from $5-1500$ μm at 1 μm resolution, calculated using the same approach and parameters for log-normal size distributions of H$_2$O. The library also includes optics files for $r_e$ values up to 20,000 μm at 500 μm resolution. This larger range is provided to accommodate sensitivity studies that probe uncertainty in

CO$_2$ snow morphology on Mars (e.g., Singh and Flanner, 2016). We assume a bulk CO$_2$ ice density of 1500 kg m$^{-3}$. The same parametric correction of $g$ for non-spherical ice particles applied to H$_2$O grains (He et al., 2017) is enabled for CO$_2$ ice. The adjustment is specific to H$_2$O and hence will be biased somewhat when applied to CO$_2$, but perhaps not by much because of their similarity in $m_r$ (Fig. 2b), which for a given shape is the main determinant of the scattering phase function, especially in weakly absorbing portions of the spectrum (e.g., Räisänen et al., 2015). We also note that some of the available shape options,

notably hexagonal habits, are unrealistic for CO$_2$ snow. Hence, the non-spherical parameterizations for CO$_2$ ice should be used with caution.

### 3.3  Light-absorbing constituents

The primary application of SNICAR has been to study the effects of LAC on snow albedo. In this section we describe the optical properties of various types of LAC that are included in the accompanying library of Mie properties. The variation in

observed LAC properties is immense and far greater than the subset of properties included here. The library presented here constitutes a sample of properties that have been used in previous works and which are broadly representative of constituent properties. We remind readers, though, that these are not "one size fits all" and that in many cases properties specific to one's application or measurements may need to be generated and applied.





A key property that governs the potency of LAC in perturbing snow albedo is the mass-absorption cross-section ($k_a$):

$$k_a = k_e(1 - \omega) \tag{15}$$

Although the LAC scattering properties are also needed in our representation (Eqs. 4–6), they generally have negligible bearing on snow albedo at typical LAC mixing ratios ($\sim 10^{-10}$ to $10^{-4}$), as snow scattering at $\lambda < 1.0\,\mu\text{m}$ is dominated by ice grains, and impurity scattering is too small to increase snow reflectance in the absorptive NIR bands of ice. Hence we only report MAC values in this section, and remind readers that scattering properties are included in the optics files. All of the properties described below are generated with Mie calculations using the standard and coated-sphere solvers of Bohren and Huffman (1983). Table 2 lists the resolved size distribution parameters and $500\,\text{nm}$ $k_a$ values for all LAC species included in the library. In addition to the resolved effective ($r_e$) and number-median ($r_n$) radii, we also list the resolved mass mean (or volume mean) radius, $r_m$:

$$r_m = \left[ \frac{\sum r^3 n(r)}{\sum n(r)} \right]^{1/3} \tag{16}$$

In the log-normal size distribution, $r_m$ is related analytically to $r_n$ as:

$$r_m = r_n \exp \left[ 1.5 \left( \ln \sigma_g \right)^2 \right] \tag{17}$$

### 3.3.1 Black carbon

Black carbon (BC) is generally defined as the strongly-absorbing component of carbonaceous aerosols, and consists largely of elemental carbon. We apply the BC optical properties described by Flanner et al. (2012). These are derived from the spectrally-resolved refractive index parameterization provided by Chang and Charalampopoulos (1990), but adjusted with linear offsets to achieve $m_r = 1.95$ and $m_i = 0.79$ at $\lambda = 550\,\text{nm}$, as recommended by Bond and Bergstrom (2006) in their comprehensive review. The parameterization we apply over $0.2 - 5.0\,\mu\text{m}$, with $\lambda$ in units of micrometers, is therefore:

$$m_r = 2.0248 + 0.1263 \ln \lambda + 0.027 (\ln \lambda)^2 + 0.0417 (\ln \lambda)^3 \tag{18}$$

$$m_i = 0.7779 + 0.1213 \ln \lambda + 0.2309 (\ln \lambda)^2 - 0.01 (\ln \lambda)^3 \tag{19}$$

We assume log-normal size distributions with $r_n = 40\,\text{nm}$ and $\sigma_g = 1.8$, intermediate values from a compilation of measurements of fresh soot (Bond et al., 2006b). The particle density is assumed to be $1270\,\text{kg}\,\text{m}^{-3}$, which achieves $k_a = 7.5\,\text{m}^2\,\text{g}^{-1}$ at $550\,\text{nm}$, conforming with the central recommendation of Bond and Bergstrom (2006). BC in snowpack has been found to have a larger size distribution than atmospheric BC in some studies (Schwarz et al., 2013), but not in others (Sinha et al., 2018). The two types of BC that can be specified are uncoated (representative of fresh, externally-mixed soot) and sulfate-coated (representative of aged, internally-mixed soot). The refractive sulfate coating, with properties from Hess et al. (1998), has an outer radius $2.15\times$ larger than the uncoated BC and produces an absorption enhancement, per unit mass of BC, of $1.5$ (Bond et al., 2006a). Various new and more sophisticated treatments of internally-mixed BC in ice have been explored (e.g.,





**Table 2.** Light-absorbing particle properties

| Particle Type | Resolved $r_e$ [µm] | Resolved $r_n$ [µm] | Resolved $r_m$ [µm] | $\sigma_g$ | Particle density $\rho$ [kg m$^{-3}$] | 500 nm $k_a$ [m$^2$ g$^{-1}$] |
|---|---|---|---|---|---|---|
| Uncoated black carbon | 0.09 | 0.04 | 0.07 | 1.8 | 1270 | 7.94 |
| Coated black carbon[a] | 0.20 | 0.09 | 0.14 | 1.8 | 1657 | 12.07 |
| Uncoated brown carbon | 0.09 | 0.04 | 0.07 | 1.8 | 1270 | 1.11 |
| Coated brown carbon[a] | 0.20 | 0.09 | 0.14 | 1.8 | 1657 | 1.65 |
| Dust Size 1 ($0.05 < r < 0.5$ µm) | 0.37 | 0.29 | 0.33 | 2.0 | 2000–2747[b] | 0.04–0.19[b] |
| Dust Size 2 ($0.5 < r < 1.25$ µm) | 0.85 | 0.70 | 0.80 | 2.0 | 2000–2747[b] | 0.04–0.16[b] |
| Dust Size 3 ($1.25 < r < 2.5$ µm) | 1.73 | 1.52 | 1.67 | 2.0 | 2000–2747[b] | 0.03–0.11[b] |
| Dust Size 4 ($2.5 < r < 5.0$ µm) | 3.28 | 2.91 | 3.18 | 2.0 | 2000–2747[b] | 0.03–0.07[b] |
| Dust Size 5 ($5.0 < r < 50$ µm) | 6.56 | 5.65 | 6.27 | 2.0 | 2000–2747[b] | 0.02–0.04[b] |
| Volc. Ash Size 1 ($0.05 < r < 0.5$ µm) | 0.32 | 0.16 | 0.25 | 2.8 | 2600 | 0.08 |
| Volc. Ash Size 2 ($0.5 < r < 1.25$ µm) | 0.85 | 0.69 | 0.79 | 2.8 | 2600 | 0.08 |
| Volc. Ash Size 3 ($1.25 < r < 2.5$ µm) | 1.78 | 1.57 | 1.72 | 2.8 | 2600 | 0.06 |
| Volc. Ash Size 4 ($2.5 < r < 5.0$ µm) | 3.48 | 3.05 | 3.35 | 2.8 | 2600 | 0.04 |
| Volc. Ash Size 5 ($5.0 < r < 50$ µm) | 8.98 | 6.07 | 7.72 | 2.8 | 2600 | 0.02 |

[a] The size parameters and densities of coated particles refer to the whole particle, whereas $k_a$ is defined with respect to core mass only. Core particle sizes are identical to their uncoated counterparts.

[b] The ranges of dust densities and $k_a$ values span the four types of dust described in Sect. 3.3.3

Liou et al., 2011; Flanner et al., 2012; He et al., 2017). All of these studies indicate that BC absorbs more solar energy, per unit mass, when it is embedded inside weakly absorbing matrices like ice, with absorption enhancement factors typically ranging from $1-2$, but with substantial variability associated with the morphology and size distributions of the inclusions, coatings, and matrices (e.g., Cappa et al., 2019). The sulfate-coated BC species therefore serves as a first-order proxy for BC that is internally-mixed in any weakly-absorbing, refractive agent (ice, sulfate, organics, etc). More accurate techniques, such as the representation of multiple inclusions of BC residing in non-spherical ice particles (He et al., 2017), should be applied when particle/inclusion habits and mixing geometry are known, but it is quite rare that these microphysical details are measured or simulated. The spectral profiles of $k_a$ for the two species of BC included in the library are shown in Fig. 3a.

### 3.3.2 Brown carbon

So-called "organic carbon" is co-emitted with BC in the combustion process, and generally consists of weakly-absorbing aerosol. Although most organic carbon has a sufficiently high $\omega$ to not appreciably influence snow albedo, the "brown carbon" subset absorbs strongly in the UV/blue and weakly in the red (Andreae and Gelencser, 2006), and is therefore of interest for snow albedo impacts (e.g., Lin et al., 2014). The optical properties of organic and brown carbon exhibit enormous diversity



**Figure 3.** Spectral mass absorption cross-sections ($k_a$) of the black carbon (a), brown carbon (b), four types of mineral dust (c–f), and volcanic ash (g) species included in the SNICAR-ADv3 optical property library, along with a small subset of the snow algae properties (h). The "Base Case" algae consists of dry cell mass fractions of 1.5% chlorophyll-$a$, 0.5% chlorophyll-$b$, 5% photoprotective carotenoids, and 0% photoreceptive carotenoids. Single-pigment scenarios with 1% mass fraction are shown in the other curves. The mean cell radius in all cases is 10 μm. Note that $k_a$ is normalized to the mass of the entire cell, which is 78% water by volume. The large $k_a$ values for snow algae at $\lambda > 2.5$ μm (exhibited in all curves) are due to absorption by water.

due to differences in solubility, size distribution, and molecular composition that are associated with combustion characteristics and the composition of the parent material (e.g., Sun et al., 2007; Laskin et al., 2015). Organic and brown carbon aerosols also originate from primary emission of biogenic particles and secondary aerosol formation from biogenic gas emissions, adding even more to the diversity of the optical properties of this class of aerosols.

Here, we provide properties for two versions of brown carbon with identical size distributions and sulfate coatings as the BC species, but derived from brown carbon imaginary refractive indices measured by Kirchstetter et al. (2004). These properties





are more absorptive than most other derivations of organic and brown carbon optical properties (e.g., Hoffer et al., 2006; Sun et al., 2007; Chen and Bond, 2010; Lack et al., 2012; Laskin et al., 2015), and thus may serve as an approximate upper bound on albedo impacts from organic/brown carbon. The native data from Kirchstetter et al. (2004) extend from $350 - 700\,\mathrm{nm}$, and

are extrapolated to $200\,\mathrm{nm}$ using a piecewise cubic hermite interpolating polynomial function (Matlab function 'pchip') that preserves the tendency towards increasing absorptivity with shorter wavelengths. Data at $\lambda > 700\,\mathrm{nm}$ are linearly tapered down to a value of $10^{-5}$ at $\lambda = 5\,\mathrm{\mu m}$. The real component of the refractive index is held constant at $1.53$. A chart showing how these properties compare with other measurements of brown carbon is shown in Figure 12 of Laskin et al. (2015). The $k_a$ values from our determination are presented in Fig. 3b, which shows UV absorptivity that is nearly as large as that of BC, but a much

sharper decline in absorptivity with wavelength. As with BC, the absorption enhancement associated with the sulfate coating is $\sim 1.5$ in the visible, but it could actually range from 1 (i.e., no enhancement) to 2 or more.

### 3.3.3 Mineral dust

Snow darkening resulting from the deposition of aeolean dust has been documented in many cases (e.g., Painter et al., 2007). Per unit mass, dust is much less absorptive than BC, but is often present in mixing ratios that are orders of magnitude greater.

Dust particles in the atmosphere and snow are typically larger, and span a wider range of sizes, than those of BC, motivating the creation of more options in SNICAR for dust particle sizes. Soil dust also exhibits substantial diversity in mineral composition, leading to variability in absorptivity. Minerals with oxidized iron, in particular, tend to strongly absorb light and thus iron content is one of the key determinants of dust absorptivity.

In SNICAR-ADv3 users can specify dust mixing ratios in each of the following five particle size bins:

1. $0.05 < r < 0.5\,\mathrm{\mu m}$

    2. $0.5 < r < 1.25\,\mathrm{\mu m}$

    3. $1.25 < r < 2.5\,\mathrm{\mu m}$

    4. $2.5 < r < 5.0\,\mathrm{\mu m}$

    5. $5.0 < r < 50\,\mathrm{\mu m}$

The smallest four of these bins match those that have been used in global aerosol transport studies (Zender et al., 2003; Mahowald et al., 2006; Scanza et al., 2015). The largest bin was added because measurements show that very large dust particles are often present in snowpack (e.g., Skiles et al., 2017), especially in patchy snow of arid environments. The dust optical properties for each of these bins are derived from Mie calculations assuming a log-normal size distribution with analytic $r_e = 1.38\,\mathrm{\mu m}$ and $\sigma_g = 2.0$ (and a mass median diameter of $3.5\,\mathrm{\mu m}$), matching the parameters used by Scanza et al. (2015) and

others, and truncated to each of the size bounds listed above. We perform Mie calculations on 1000 log-spaced particle radii within each size bin. The resolved $r_e$, $r_n$, and $r_m$ values for each size bin are listed in Table 2.

We provide optical properties for four compositions of dust that have been used in previous dust-on-snow studies, representative of properties from different geographies and derived using diverse techniques and data:



1. *Saharan dust*: For this species we apply mineral fractions from the "central hematite" scenario presented by Balkanski et al. (2007), which produced good agreement with AERONET measurements in regions strongly affected by Saharan dust. These mineral volume fractions are: 31.5% illite, 24% kaolinite, 23% montmorillonite, 14% quartz, 6% calcite, and 1.5% hematite. We apply the Bruggeman mixing approximation to derive a particle-mean dielectric constant ($\epsilon = M^2$) from that of each of the $i$ minerals ($\epsilon_i$) and mineral volume fractions ($V_i$). Specifically, we use an iterative secant method (Flanner et al., 2012) to find $\epsilon$ such that:

$$\left| \sum_i V_i \frac{\epsilon_i - \epsilon}{\epsilon_i + 2\epsilon} \right| < \varepsilon, \qquad \text{where } \varepsilon = 10^{-12} \tag{20}$$

We then use the particle-average refractive indices in Mie calculations with the size distribution parameters listed above. The spectrally-resolved mineral refractive indices we use are essentially those tabulated in the supplement of Scanza et al. (2015), which originate from several sources (Egan et al., 1975; Egan and Hilgeman, 1979; Querry, 1987; Long et al., 1993; Rothman et al., 1998, A.H.M.J. Triaud, personal communication, 2005). We also assume the same mineral densities as Scanza et al. (2015) (units of $\mathrm{kg\,m^{-3}}$): illite: 2750, kaolinite: 2600, montmorillonite: 2350, quartz: 2660, calcite: 2710, hematite: 5260, resulting in a particle-average density of $2645\,\mathrm{kg\,m^{-3}}$.

2. *Colorado dust*: Refractive indices of dust collected from snow in the San Juan Mountains of Colorado were determined from Mie inversions and measurements of the spectral albedo of optically-thick dust samples of known particle size distribution (Skiles et al., 2017). This technique finds the $m_i$ value at each wavelength that optimize the comparison between measured and modeled dust spectral albedo. The real component ($m_r$) was assumed to be $1.525$ for the inversions. Due to an apparent inconsistency between the $m_i$ values reported by Skiles et al. (2017) and their measured dust particle size distributions, we we-ran the inversion to produce revised $m_i$ values, assuming a lognormal size distribution with the same resolved effective radius ($r_e = 2.3\,\mu\mathrm{m}$) of the measurements. The revised $m_i$ values are larger than those from the original inversion. We extrapolated $m_i$ to wavelengths shorter than $350\,\mathrm{nm}$ with a shape-preserving polynomial (Matlab function 'pchip'). At $\lambda > 2.5\,\mu\mathrm{m}$, $m_i$ is assumed to equal $0.00165$, the $2.5\,\mu\mathrm{m}$ value derived from our inversion. We assume a dust density of $2600\,\mathrm{kg\,m^{-3}}$ and $\sigma_g = 1.32$, consistent with Skiles et al. (2017). This dataset has the advantage of deriving from optical measurements of bulk dust samples, and is not vulnerable to uncertainties about the mixing state, mixing ratios, and refractive indices of individual minerals, as occurs in the modeling of dielectric properties for the other types of dust we provide. This technique may, however, be susceptible to errors in albedo measurements, unaccounted light transmission, the assumption of spherical particles, and uncertainty in $m_r$.

3. *Greenland dust*: Our approach for calculating Greenland dust properties follows that used for the Saharan dust, but with mineral composition from Polashenski et al. (2015). These mineral fractions were estimated from linear mixing model calculations using measurements of aluminum, iron, non-sea salt calcium, water insoluble potassium, and arsenic in snow samples collected across Greenland. Three scenarios of hematite fraction were derived by Polashenski et al. (2015), based on uncertainties in mineral derivations and previous studies of Greenland dust. The central scenario, applied here, has mineral mass (volume) fractions of: 31% (31%) illite, 14.4% (15.2%) kaolinite, 6.2% (7.3%) montmorillonite, 8.1%





(8.3%) quartz, 34.9% (35.4%) calcite, and 5.4% (2.8%) hematite. The particle mean density is $2747\,\mathrm{kg\,m^{-3}}$. This mineral composition is consistent with ice core analyses from Greenland and other studies suggesting that much of the remote dust depositing on Greenland originates from Asia.

4. *Martian dust*: Dust is a prominent feature of the "Red Planet" and contributes to large reductions in the albedo of its polar ice caps. With the addition of $CO_2$ ice to SNICAR, new capabilities exist to model the impacts of dust on the albedo of extraterrestrial ice surfaces (Singh and Flanner, 2016; Singh et al., 2018). The refractive indices used to generate our Martian dust are derived from spectral measurements from the Mars Reconnaissance Orbiter (Wolff et al., 2009, 2010, Mike Wolff, personal communication). We apply the shortest-wavelength ($\lambda = 263\,\mathrm{nm}$) value from this dataset

constantly across the short-spectrum ($\lambda = 200 - 263\,\mathrm{nm}$) portion of SNICAR. We also assume a particle density of $2000\,\mathrm{kg\,m^{-3}}$ in deriving the optical properties, consistent with Singh and Flanner (2016).

Spectral mass-absorption cross-sections ($k_a$) for these dust types and size bins are shown in Fig. 3c–f. Per unit mass, small dust particles are more effective absorbers than larger ones. All dust species are more absorptive in the blue than red, and smaller particles exhibit a stronger spectral dependency in absorption. The jagged feature near $350 - 400\,\mathrm{nm}$ in the Saharan

and Greenland dust originates from the hematite refractive indices. Mie non-linearities are also apparent. For example, the finest Greenland dust is the most absorptive of all types, which is not surprising given its relatively large hematite content, but the mid-size Martian dust is more absorptive than its counterparts. Overall, Colorado dust has lower visible $m_i$ than the other dust species included in our library, but is an important component of our collection because of the diversity in dust absorptivity that it indicates, the importance of southwest Colorado for dust-on-snow studies (e.g., Skiles and Painter, 2019),

and the unique way in which the properties were determined. Singh and Flanner (2016) and Singh et al. (2018) applied a gamma size distribution of dust, with $r_e = 1.5\,\mathrm{\mu m}$ and effective variance of 0.3 (Wolff et al., 2006), yielding very similar $k_a$ as that of the central size bin (shown in Fig. 3f for reference). These properties are also included in the accompanying library.

### 3.3.4   Volcanic ash

Deposition of volcanic ash on snow, while highly episodic, can substantially lower albedo (e.g., Conway et al., 1996; Young

et al., 2014; Gelman Constantin et al., 2020). We apply ash refractive indices from the "central forcing" scenario presented by Flanner et al. (2014) (their Figure 2). These are derived from a variety of measurements following the 2010 Eyjafjallajökull eruption, including chemical and optical measurements of ash in the atmosphere and snow (Schumann et al., 2011; Bukowiecki et al., 2011), and inversions of sun photometer, ground-based lidar, and particle soot absorption photometer (PSAP) measurements (Hervo et al., 2012; Toledano et al., 2012; Weinzierl et al., 2012). These central scenario $m_i$ values are very similar to

those of ash collected from the 1980 Mount St. Helens eruption (Patterson, 1981). The $500\,\mathrm{nm}$ $m_i$ value we apply is 0.0044. Flanner et al. (2014) presented low and high scenarios of ash absorptivity that ranged from 0.0014 to 0.014, reflecting both measurement uncertainty and real variability in ash absorptivity. The $m_r$ component in this dataset is 1.54 at $\lambda \leq 0.55\,\mathrm{\mu m}$, sloping down to 1.47 at $5.0\,\mathrm{\mu m}$. We calculate ash optical properties in the same five size bins used for dust, and with log-normal $\sigma_g = 2.8$ and effective radius of $r_e = 2.29\,\mathrm{\mu m}$, which matches the effective radius of global-mean ash deposition simulated





across 25 size bins by Stohl et al. (2011). We assume a particle density of $2600\,\mathrm{kg\,m^{-3}}$. Ash particles are often highly aspherical, calling into question the appropriateness of Mie calculations. Flanner et al. (2014) performed a number of sensitivity studies using $T$-Matrix calculations (Mishchenko and Travis, 1998) and found that, although the scattering properties differ substantially between equal-volume spheres and non-sphere ash particles, the $k_a$ values differed by at most 16%, leading to the conclusion that particle shape contributed only second-order uncertainty to the estimation of $k_a$, the key optical property for snow impurity albedo studies. The volcanic ash $k_a$ values are depicted in Fig. 3g, which shows lower absorptivity than all types of dust presented except for Colorado dust.

### 3.3.5 Snow algae

Algae blooms occur in seasonal snowpack and in the ablation zones of ice sheets and glaciers (e.g., Takeuchi et al., 2001; Painter et al., 2001; Cook et al., 2017). Snow and glacier algae exhibit enormous genetic and phenotypic diversity (e.g., Lutz et al., 2014), and are thus difficult to represent using traditional physical models. Our goal here is to provide a first-order representation that accounts for light-absorption by some of the key pigments present in algae, utilizing the approach introduced by Cook et al. (2017). Given the focus of this study on snow albedo modeling and the pending development of a separate glacier albedo model (*Whicker et al.*, in prep.), we focus exclusively on snow algae, which are often reddish or greenish and can be detected remotely because of their unique spectral characteristics (e.g., Painter et al., 2001; Wang et al., 2018). Our approach should be considered an experimental work-in-progress that will evolve with improved observational constraints on the nature and determinants of snow algal absorption, as well as improved modeling techniques for representing heterogeneous and irregularly-shaped media like algal cells (Cook et al., 2020).

We utilize the spectral mass absorption cross-sections of various algal light-absorbing pigments from Dauchet et al. (2015) to derive pigment $m_i$ values and assume a constant $m_r = 1.5$ for all pigments (Pottier et al., 2005; Cook et al., 2017). We apply the Bruggeman approximation described earlier to mix these properties (and those of water) in various proportions and then conduct Mie calculations using the cell-average refractive indices. We assume a cell water volume fraction of $0.78$ (Pottier et al., 2005; Dauchet et al., 2015) and consider various dry cell mass fractions of the following four pigments:

1. Chlorophyll-$a$ [0, 0.005, 0.01, 0.015, 0.02, 0.025, 0.03]

2. Chlorophyll-$b$ [0, 0.005, 0.01, 0.015, 0.02, 0.025, 0.03]

3. Photoprotective carotenoids [0, 0.01, 0.02, . . . , 0.15]

4. Photosynthetic carotenoids [0, 0.01, 0.02, . . . , 0.15]

The numbers in brackets indicate the discretized dry cell mass fractions for which we created optical properties. Chlorophyll-$a$ pigments are present in all photosynthetic snow algae, and the accessory chlorophyll-$b$ pigment is also common (e.g., Remias et al., 2005). The carotenoid properties refer to generic classes of pigments, with the photoprotective carotenoids particularly relevant, as algae often produce these pigments to protect themselves from UV radiation and excessive solar heating (e.g., Remias et al., 2005). The dry cellular matter unaccounted for with the pigments is assumed to have constant $M = 1.5 + 0i$



(i.e., to be non-absorbing) (Pottier et al., 2005; Cook et al., 2017). All dry matter is assumed to have a density of $1400\,\mathrm{kg\,m^{-3}}$ (Dauchet et al., 2015), yielding a mean cell density of $\rho_{alg} = 1088\,\mathrm{kg\,m^{-3}}$. The $m_i$ properties from Dauchet et al. (2015) are only defined down to $\lambda = 350\,\mathrm{nm}$, and the use of shape-preserving extrapolating functions is suspect due to the sharp absorption features of pigments. We therefore take a somewhat conservative approach of assuming that $m_i$ at $\lambda = 200\,\mathrm{nm}$ (our lower model boundary) is equal to half of the published values at $\lambda = 350\,\mathrm{nm}$, and interpolate linearly between $200\,\mathrm{nm}$ and $350\,\mathrm{nm}$.

We assume Gaussian size distributions of algae cells, instead of log-normal distributions as applied for all other LAC. The standard deviation ($\sigma$) of the distribution is assumed to be 10% of the mean cell radius ($r_{alg}$), and for each mean radius we perform Mie calculations over 200 equally-spaced size bins ranging across $r_{alg} \pm 4\sigma$. The mean cell radii for which we generated properties are: 1, 2, 5, 10, 15, 20, 25, 30, 40, and $50\,\mu\mathrm{m}$, with $r_{alg} = 10\,\mu\mathrm{m}$ set as the default. Remias et al. (2005) report a mean diameter of $14.9 \pm 5.7$ for quasi-spherical cells of *C. nivalis*, one of the most prevalent species of algae in mountain and polar snowpack.

We generate a library of algae optical properties, building on the "BioSNICAR" library developed by Cook et al. (2017), with five dimensions: mean cell radius and dry cell mass fractions of each of the four pigments listed above. In total, the sampling and discretization over these five dimensions results in 125,440 unique combinations of Mie properties, and the algae component of the optics library therefore occupies the majority (8.2 GB) of its disk space. Users specify an algae cell number concentration ($\mathcal{N}$) in units of cells per mL of meltwater, which is consistent with how the microbiology community measures and reports cell concentrations. Internally, however, SNICAR treats all LAC in terms of mass mixing ratios and mass-specific optical properties (e.g., Eqs. 1–3). We therefore convert the user-specified cell number concentration to a mass mixing ratio ($q_{alg}$) using Gaussian statistics of the size distributions applied in Mie calculations, i.e.:

$$q_{alg} = 10^{-15} \mathcal{N} \rho_{alg} \frac{4}{3} \pi \left( r_{alg}^3 + 3 r_{alg} \sigma^2 \right) \tag{21}$$

where the factor of $10^{-15}$ is needed with $r$ in $\mu\mathrm{m}$, $\mathcal{N}$ in $\#\,\mathrm{mL^{-1}}$, and $\rho$ in $\mathrm{kg\,m^{-3}}$.

Fig. 3h depicts $k_a$ values for a very small subset of this library, with one example for our algae "base case" pigment fractions and cell size (Table 3), and four end-member examples with only one type of pigment present to highlight the spectral features of individual pigments. The base case dry cell pigment mass fractions are 0.015 chlorophyll-$a$, 0.005 chlorophyll-$b$, 0.05 photoprotective carotenoids, and no photoreceptive carotenoids. This is similar to the "medium" scenario from Cook et al. (2017), but with a chlorophyll-$b$ / chlorophyll-$a$ ratio (0.33) in-line with measurements of *C. nivalis* from Remias et al. (2005). The base case cell radius, and that of all curves shown in Fig. 3h, is $10\,\mu\mathrm{m}$, slightly larger than that measured by Remias et al. (2005) for *C. nivalis*, and slightly smaller than the medium scenario given by Cook et al. (2017). Sharp absorption features are apparent in Fig. 3h, including the chlorophyll-$a$ absorption peaks near 430 and $670\,\mathrm{nm}$ that are exploited for remote sensing retrievals (Wang et al., 2018), and the more closely spaced chlorophyll-$b$ peaks. These become partially obscured when other pigments are present, as seen in the base case absorption profile, and can become even more obscured when other broad spectrum photoprotective pigments are present (Cook et al., 2020). Note that $k_a$ shown in Fig. 3h is that of the entire cell,





**Table 3.** Model base case parameters for sensitivity studies and default parameters for the web-based model

| Model feature | State |
|---|---|
| Atmospheric state for surface irradiance | Mid-latitude winter |
| Direct or diffuse incident irradiance | Direct |
| Solar zenith angle (SZA) | 60° |
| Ice refractive index data | Merged Picard et al. (2016) and Warren and Brandt (2008) |
| Snowpack thickness | 100 m (i.e., optically semi-infinite) |
| Snowpack density* | 200 kg m$^{-3}$ |
| Albedo of underlying ground* | 0.25 |
| Snow grain shape | Hexagonal plates |
| Snow grain size | 100 μm (fine-grain base case) |
| | 1000 μm (coarse-grain base case) |
| Mixing ratios of all LAC | Zero |
| Base case algal cell parameters for snow algae studies | |
| Cell radius ($r_{alg}$) | 10 μm |
| Size distribution standard deviation | 1 μm (10% of $r_{alg}$, in general) |
| Dry cell mass fraction of chlorophyll-$a$ | 1.5% |
| Dry cell mass fraction of chlorophyll-$b$ | 0.5% |
| Dry cell mass fraction of photoprotective carotenoids | 5.0% |
| Dry cell mass fraction of photosynthetic carotenoids | 0% |

*Snowpack density and underlying ground albedo have no impact on simulated albedo when the snowpack is semi-infinite.

which is 78% water by volume. The absorption peak near $\lambda = 3\,\mu m$ seen in all cases is due to cell water. Normalization of cell $k_a$ to just the dry cell mass, as is sometimes reported, would produce $3.53\times$ larger values.

The techniques we utilize to represent algae are imperfect. First, the Bruggeman approximation assumes perfectly homogeneous mixing, whereas pigments have discrete sizes, shapes, and positions within the cell. In cases where such features of the cell are known, techniques like the dynamic effective medium approximation (Chýlek and Srivastava, 1983), discrete dipole approximation (Draine and Flatau, 1994), or geometric-optics surface-wave approach (Liou et al., 2011) might be more appropriate. Second, Mie approximations assume spheres. While glacier algae are often filamentous and generally highly non-spherical (e.g. Yallop et al., 2012) and more appropriately represented with geometric optics or other techniques (Cook et al., 2020), some of the most common types of snow algae like *C. nivalis* actually are quasi-spherical (Remias et al., 2005; Lutz et al., 2014), suggesting that Mie approximations may be reasonable for snow algae. Third, other types of pigments are present in algae, particularly specific types of carotenoids whose spectral properties may differ from those applied here (e.g., Remias et al., 2012; Williamson et al., 2020). Our technique of generating an offline library of optical properties that comprehensively





spans the parameter space precludes the use of many types of pigments. A more efficient technique may be to generate algae optical properties "on the fly" with albedo calculations, using Mie Theory or other parametric techniques. We leave it to future studies, however, for deeper dives into the representation of snow algae.

## 4  Model Sensitivities

In this section we briefly explore the sensitivity of modeled spectral and broadband albedo to some of the model parameters and design features described above. Due to the immensity of the parameter space, this analysis is not comprehensive. We define a base state of model features, listed in Table 3, from which we perturb individual parameters. Because effective snow grain size ($r_e$) is a dominant source of variability in snow albedo across Earth's snowpacks, we explore many of the sensitivities under both a coarse-grain (aged snow) base case with $r_e = 1000\,\mu m$ and the default fine-grain (fresh snow) base case with $r_e = 100\,\mu m$. The base parameters listed in Table 3 are also the default parameters used in the web-based model.

### 4.1  Pure snow

Surface spectral irradiance has no impact on spectral albedo, but influences broadband albedo ($\overline{\alpha}$) as shown in Eq. 7. Table 1 lists the base case $\overline{\alpha}$ for all atmospheric profiles. Under cloudy skies a higher proportion of the surface irradiance lies in the visible spectrum, where snow is most reflective, and hence $\overline{\alpha}$ is larger under cloudy conditions (e.g., Wiscombe and Warren, 1980). Other factors equal, clear-sky $\overline{\alpha}$ also increases with increasing water vapor content due to near-infrared absorption in the atmosphere, as seen by comparing $\overline{\alpha}$ under winter and summer atmospheric profiles (Table 1).

Fig. 4 depicts modeled spectral and broadband albedo sensitivities to ice refractive index dataset (panels a, b, and d), snow $r_e$ (panels c and d), SZA with fine and coarse grains (panels e and f), and grain shape with fine and coarse grains (panels g and h). Sensitivity to the choice of $H_2O$ ice refractive index is negligible in most of the spectrum. Even in the UV/blue spectrum $\alpha$ differs by $< 0.01$ with $r_e = 100\,\mu m$ between the two most recent and most reliable datasets (Warren and Brandt, 2008; Picard et al., 2016), but these differences increase to $0.025$ with $r_e = 1000\,\mu m$. Differences are even larger between the Warren (1984) and Warren and Brandt (2008) datasets but the former is obsolete. Ice absorption is so weak in the Warren and Brandt (2008) dataset that UV/blue snow albedo is nearly 1.0 regardless of grain size (Figs. 4a and 4b). $CO_2$ snow is more reflective than $H_2O$ snow, except in the highly absorptive NIR bands of $CO_2$ (Fig. 4d, Singh and Flanner, 2016). Snow $r_e$ (or SSA) is well-known to be one of the primary controls on snow albedo (e.g., Warren and Wiscombe, 1980), as shown in Fig. 4c. The influence is primarily in the NIR part of the spectrum, and $\overline{\alpha}$ ranges from $0.848 - 0.747$ for $r_e = 100 - 1000\,\mu m$ in our base case. It is also well-known that clear-sky albedo increases with increasing SZA, as manifested in Figs. 4e and 4f. The range of $\overline{\alpha}$ across SZA of $0 - 85°$ is $0.819 - 0.862$ in the fine-grain base case (Fig. 4e), and is $0.707 - 0.758$ in the coarse-grain base case (Fig. 4f). At large SZA, however, the albedo dependency is weakened due to a red-shift in surface spectral irradiance that occurs with Rayleigh scattering (van Dalum et al., 2020), which by itself decreases $\overline{\alpha}$ and thus counteracts the tendency towards larger $\overline{\alpha}$ due merely to increasing incidence angle. We see noticeable divergence in $\overline{\alpha}$ between simulations with and without SZA-



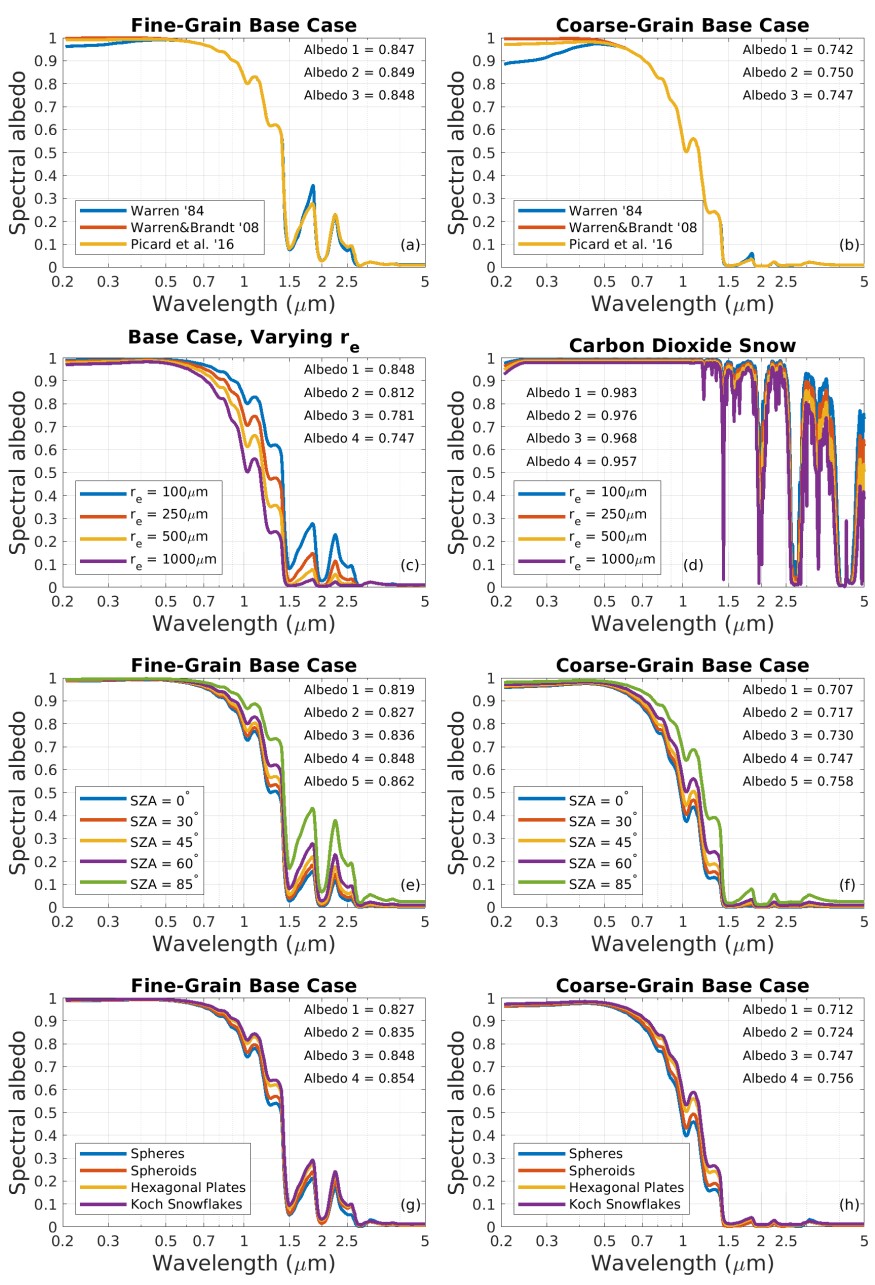

**Figure 4.** Albedo variations of optically semi-infinite, impurity-free snow in SNICAR-ADv3. Variations are shown for different ice refractive index datasets (a,b,d), ice particle effective grain size (c,d), solar zenith angle (e,f), and ice particle shape (g,h). Except for the parameter being varied, all snowpack properties are those of the fine-grain ($r_e = 100\,\mu m$) or coarse-grain ($r_e = 1000\,\mu m$) base case defined in Table 3. Solar broadband albedos for each curve are also printed on the figures, with order corresponding to that of the legend.





dependent surface irradiance at SZA $>\sim 65°$ due to more complex spectral shifts associated with long path-length gaseous
absorption and Rayleigh scattering.

Snow grain shape and morphology have gained renewed attention for their importance in snow radiative transfer (e.g.,
Kokhanovsky and Zege, 2004; Liou et al., 2011; Libois et al., 2013; Räisänen et al., 2015; Dang et al., 2016; He et al.,
2017). Figs. 4g and 4h show that non-spherical ice grains are associated with higher albedo than equal-SSA spherical grains,
particularly in the NIR. This is due to differences in the scattering asymmetry parameter ($g$), which is actually the only optical
parameter that changes with grain shape in our parameterization (He et al., 2017). Non-spherical ice grains scatter less strongly

in the forward direction than spheres (e.g., Fu, 2007; Libois et al., 2013; Räisänen et al., 2015; Dang et al., 2016), which
decreases the penetration depth of radiation in snow and increases the probability that photons will re-direct out of the top of
the snowpack, thus increasing albedo when all other factors are equal. For the three non-spherical grain shapes and default
aspect ratios considered in SNICAR-ADv3, Koch snowflakes have the smallest $g$ and largest $\overline{\alpha}$ relative to spheres, followed
by hexagonal plates and spheroids. The range in $\overline{\alpha}$ across all four shapes is $0.827 - 0.854$ in the fine-grain case (Fig. 4g) and

is $0.712 - 0.756$ in the coarse-grain case (Fig. 7h). Thus, as with most other factors, the influence of grain shape is greater
in larger-grain snow. Because it is now clear that spherical ice grains have unrealistically large values of $g$, we assign a non-
spherical grain shape, hexagonal plates, as the default grain shape in SNICAR-ADv3, though with the recognition that grain
shape is enormously diverse in real snowpack.

Fig. 5 depicts the dependency of albedo on snow thickness, snow grain size, and grain shape. Light penetrates most deeply in

coarse-grain, spherical, low-density snow, translating into a greater snow thickness needed to completely mask the underlying
ground, or to achieve "semi-infinite" thickness. In the fine-grain base case (Fig. 5a), light penetration is shallow and $\overline{\alpha}$ is high
(0.762) even with a $1\,\mathrm{cm}$ thick snowpack overlying a relatively dark ($\alpha = 0.25$) surface. In the coarse-grain base case (Fig.
5b), however, we see stronger dependency of $\overline{\alpha}$ on snowpack thickness. Even with a $20\,\mathrm{cm}$ thick snowpack, $\overline{\alpha}$ is 0.026 less
than its semi-infinite value. Finally, we see the importance of grain shape once again in Fig. 5c, which shows $\overline{\alpha}$ ranging from

$0.610 - 0.705$ for different shapes with $r_e = 1000\,\mathrm{\mu m}$ and a snowpack thickness of $10\,\mathrm{cm}$. These ranges narrow with smaller
$r_e$ and thicker snowpack.

## 4.2   Snow with LAC

Here, we briefly describe some of the modeled albedo sensitivities to LAC in SNICAR-ADv3. Sensitivities to BC and the
variant of brown carbon we apply are shown in Fig. 6. Part-per-billion mixing ratios of both species can substantially reduce

snow albedo, and more so in coarse-grain than fine-grain snow. In the base case, the reduction in $\overline{\alpha}$ caused by 10 ppb of uncoated
BC increases from 0.0023 to 0.0070 with an increase in $r_e$ from $100\,\mathrm{\mu m}$ to $1000\,\mathrm{\mu m}$. The ratio of coarse-grain $\overline{\alpha}$ reduction to
fine-grain $\overline{\alpha}$ reduction, greater than 3 at small $q_{BC}$, decreases slightly to 2.7 with $q_{BC} = 1000$ ppb. The albedo spectra show
that BC impacts snow albedo rather uniformly across the visible spectrum, and with some impact up to $\lambda \approx 1.0\,\mathrm{\mu m}$, whereas
brown carbon exerts a much greater influence at UV/blue wavelengths and has almost no impact at $\lambda > 0.7\,\mathrm{\mu m}$. Broadband

albedo reduction from brown carbon is $0.24 - 0.30\times$ that from equal mixing ratios of BC over the parameter ranges shown
in Fig. 6, though we again remind readers that the $k_a$ values we apply for brown carbon are higher than most estimates.



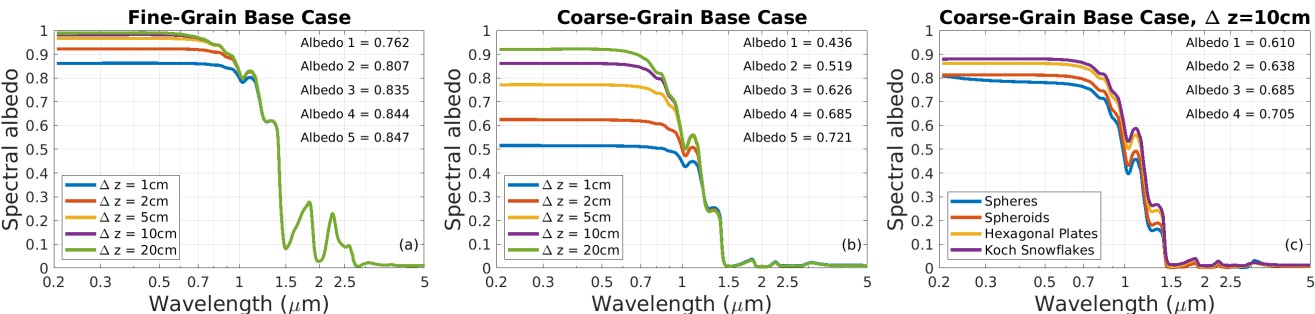

**Figure 5.** Albedo variations of thin, impurity-free snow in SNICAR-ADv3. Spectral albedos are shown for varying snowpack thickness in the fine-grain (a) and coarse-grain (b) base cases, and for different ice grain shapes with a fixed snowpack thickness (c). The order of broadband albedos printed on each figure corresponds with the legend order.

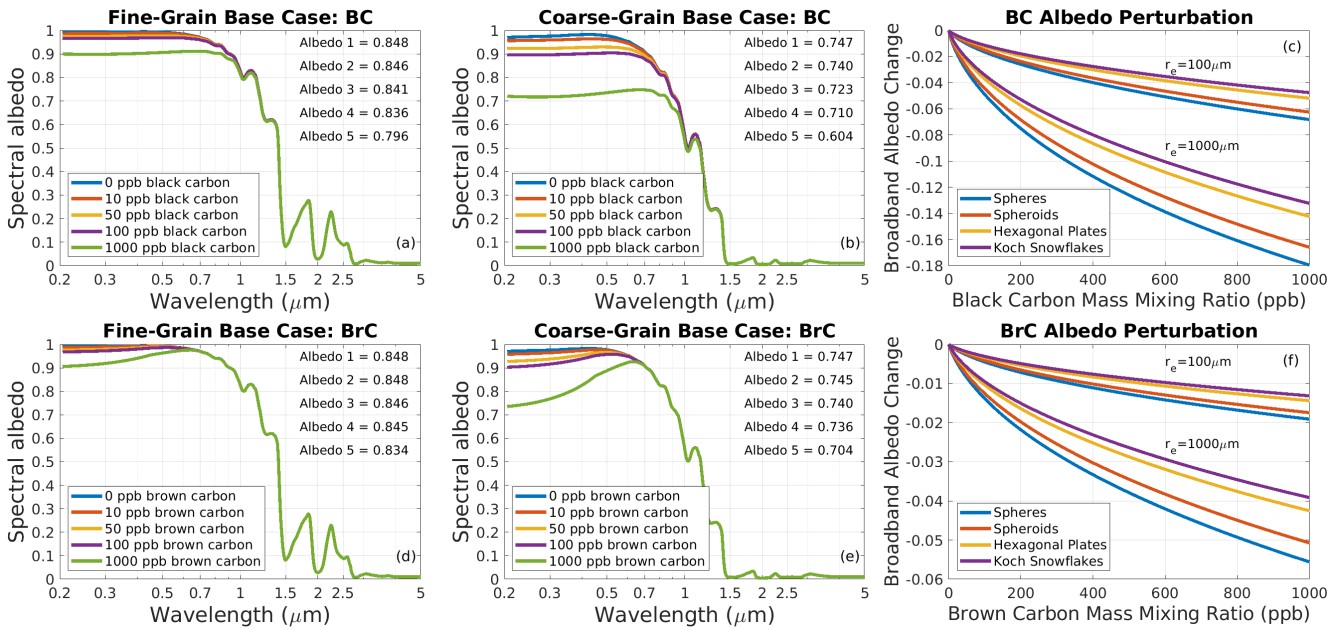

**Figure 6.** Albedo variations caused by uncoated black carbon (top row) and brown carbon (bottom row). Spectral albedo perturbations for fine-grain (a and d) and coarse-grain (b and e) base cases are shown, as well as broadband albedo perturbations as a function of impurity content (c and f). The order of broadband albedos printed on each figure corresponds with the legend order.

The sulfate-coated versions of BC and brown carbon, with larger $k_a$, drive larger albedo reduction per unit mass of particle. Across the $r_e$ and mixing ratio ranges shown in Fig. 6, the coated particles reduce $\overline{\alpha}$ by $1.2 - 1.4\times$ more than their uncoated counterparts (not shown). We suggest that these coated species may serve as reasonable proxies for internally-mixed particles in ice, as the albedo reduction enhancement was shown to be similar $(1.3 - 1.6\times)$ for BC residing inside snow grains compared with externally-mixed BC (He et al., 2018).





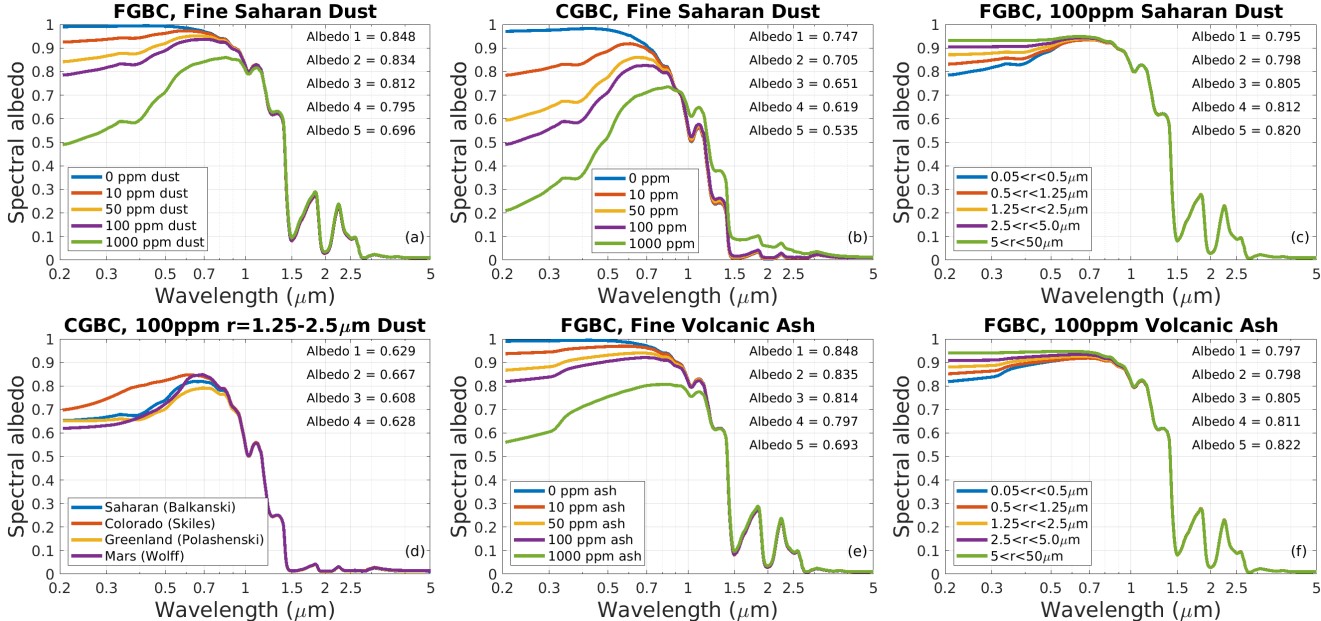

**Figure 7.** Albedo variations caused by mineral dust (a–d) and volcanic ash (e–f). Spectral albedo perturbations are shown for fine grain (a) and coarse grain (b) base cases with different mixing ratios of the smallest size bin of Saharan dust, for different particle sizes of Saharan dust with a mixing ratio of 100 ppm in the fine-grain base case (c), and for different types of dust in the coarse-grain base case (CGBC), with an intermediate particle size ($1.25 < r < 2.5\,\mu m$) and mixing ratio of 100 ppm. Albedo variations for volcanic ash are shown for different mixing ratios (e) and particle sizes (f) in the fine-grain base case. The order of broadband albedos printed on each figure corresponds with the legend order.

An important consequence of shallower light penetration in snow with non-spherical ice grains is that the albedo perturbation from LAC is reduced (Dang et al., 2016), as fewer LAC are exposed to light. Figs. 6c and 6f show the broadband albedo reduction from BC and brown carbon in snow with different ice grain shapes. The use of hexagonal plates as the default grain shape in SNICAR-ADv3 instead of spheres reduces the simulated albedo impact from BC by $\sim 24\%$, with only slight variability associated with grain size or BC amount. The BC albedo perturbation in snow composed of spheroids and Koch snowflakes is $\sim 9\%$ and $\sim 31\%$ less, respectively, than in snow composed of spheres, and these ratios are very similar with brown carbon (Fig. 6f). Because this variability is associated with the penetration depth of light in snow, it also highlights the importance of accurately representing the vertical distribution of LAC, which can be modeled with multi-layer configurations of SNICAR-ADv3.

Albedo changes induced by dust and volcanic ash are shown in Fig. 7. Mixing ratios up to $1000\,\mathrm{ppm}$ of fine ($0.05 < r < 0.5\,\mu m$) Saharan dust reduce albedo by up to $0.15$ in fine-grain snow (Fig. 7a) and $0.21$ in coarse-grain snow (Fig. 7b). This type of dust (like most dusts) gives the snow a reddish/brown appearance by disproportionately reducing blue albedo. With a mass mixing ratio of $100\,\mathrm{ppm}$, we see that the albedo reduction from Saharan dust ranges from $0.028 - 0.053$ in the fine-grain base



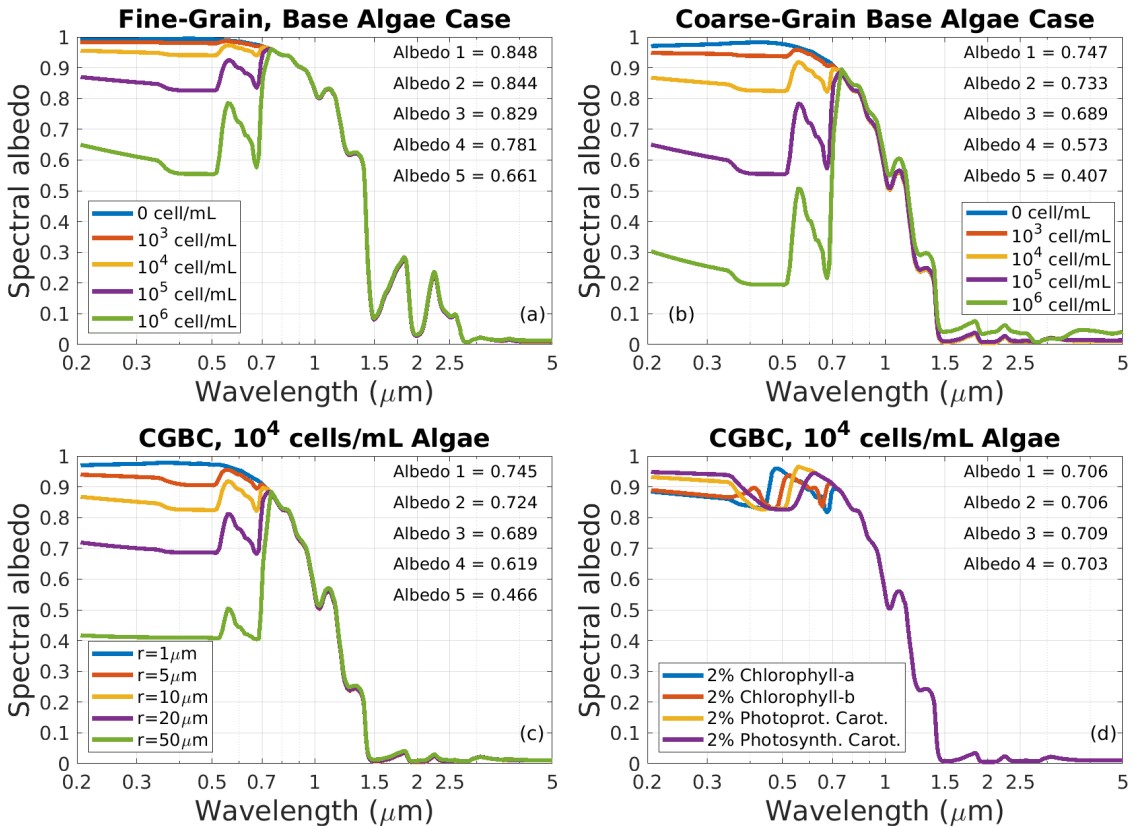

**Figure 8.** Albedo variations caused by snow algae in SNICAR-ADv3 with default pigment fractions (Table 3) in the fine-grain base case (a) and coarse-grain base case (b). Dependencies on algal cell radius (c) and individual pigment types (d) are also shown for the coarse-grain base case (CGBC) with a cell number concentration of $10^4$ mL$^{-1}$. The order of broadband albedos printed on each figure corresponds with the legend order.

case, depending on the size bin of particles, with fine dust having the greatest mass-normalized influence (Fig. 7c). For a given set of snow conditions and particle size, dust type contributes to modest variability in albedo change (Fig. 7d), with Martian dust the most absorptive and Colorado dust the least. Other observed mixtures of dust would produce greater variability (Scanza et al., 2015), however, and our representation of dust may evolve in the future to accommodate this diversity. The volcanic ash from Eyjafjallajökull has a similar impact on $\overline{\alpha}$ as Saharan dust, though via a more uniform spectral influence that gives the snow more of a greyish/brownish appearance (Fig. 7e). As with dust, larger volcanic ash particles reduce snow albedo less per unit mass than smaller ones (Fig. 7f), and the disparity widens with increasing snow grain size (not shown).

Snow algae can introduce unique spectral characteristics to snow albedo (Fig. 8). With the base case snow and algae properties (Table 3), algae number concentrations of $10^3 - 10^6$ mL$^{-1}$ reduce snow albedo by $0.004 - 0.184$ (Fig. 8a), increasing to $0.014 - 0.336$ in the coarse grain base case, which for algae is a much more realistic scenario because they generally only proliferate in snow that has been wet for an extended period of time. Observed cell concentrations in snow and glacier algal





blooms are $\sim 10^3 - 10^5 \,\mathrm{mL}^{-1}$ (Painter et al., 2001; Yallop et al., 2012; Lutz et al., 2014; Cook et al., 2020; Williamson et al., 2020), though measurements are often made from a thicker layer of snow than that in which most of the algae reside, implying higher concentrations at the very top of the snow. Simulated albedo impacts at $0.2 < \lambda < 0.35$ should be discounted due to our lack of pigment $m_i$ data in this spectral range (Sect. 3.3.5). Unsurprisingly, larger cells cause greater reduction in albedo when

comparing fixed number concentrations (Fig. 8b). Normalizing to mass, however, as we do for other LAC, would show that smaller algae absorb more per unit mass (not shown). Fig. 8d shows snow spectra for idealized heavy-load cases where only one type of pigment is present in the cell (as in Fig. 3h), included simply to highlight that unique spectral features can arise when the pigments are blended in different proportions. The two chlorophyll pigments create reflectance peaks at slightly different wavelengths near $\lambda = 500\,\mathrm{nm}$, for example, and some combinations of carotenoids and chlorophyll cause distinctively

green or red snow, perhaps similar to the "watermelon snow" that is caused by *C. nivalis* and often observed by backcountry enthusiasts. Algae, even more so than other LAC, tend to be concentrated near the very top (e.g., $< 1\,\mathrm{cm}$) of the snow, suggesting that use of the multi-layer model with thin layers near the surface is necessary to properly resolve the influence of algae on albedo (Cook et al., 2017, 2020). We also remind readers that this should be considered an experimental technique to represent snow algae, and that more observational and model development work is needed on this topic, including the explo-

ration of techniques such as the SNICAR-GO formulation (Cook et al., 2020; Williamson et al., 2020) for representing highly non-spherical glacier algae.

## 5 Model Evaluation

We now evaluate SNICAR-ADv3 against published measurements of snow spectral albedo. Perfect constraint of the model parameter space through observations is not achievable due to the number of options and ability to specify layers of any

thickness. The vertical resolution of measurements is particularly important for model evaluation across the spectrum, as grain size and shape of only the top sub-millimeter of snow determine albedo across most of the NIR spectrum, whereas the vertical profiles of grain characteristics and impurities throughout the top $10+\,\mathrm{cm}$ of snow determine visible albedo. In the examples below, we include model scenarios limited to the resolution and extent of measurements, as well as additional scenarios that apply speculative but reasonable assumptions. Unless otherwise stated, model base case parameters are applied.

We utilize data from the following studies:

- Grenfell et al. (1994): This study includes measurements of ultra-clean snow at the South Pole and Vostok, Antarctica in 1985–1986 and 1990–1991, respectively, across a spectral range of $0.31 - 2.5\,\mathrm{\mu m}$. We apply the diffuse spectral albedo and standard deviations listed in Table 6 of that study, which represent average data from the two sites and show a narrow range of variability. In the base model scenario, we apply the vertical profiles of grain size and snow density reported in

Tables 3 and 5 of that study. The measurements of grain size were conducted visually with a reported accuracy of about $25\,\mathrm{\mu m}$. We also assume a sulfate-coated BC mass mixing ratio of $0.3\,\mathrm{ppb}$, as reported, which has negligible impact on albedo.





- Hudson et al. (2006): This study focuses on snow bidirectional reflectance, but also includes spectral albedo measurements under diffuse sky conditions from Dome C, Antarctica, across a spectral range of $0.35 - 2.5\,\mu m$. We use the data shown in Figure 6 of that study, which represent an average of five measured spectra. Vertical grain size measurements were not included, but the authors state that the surface snow was composed of grains with radii of 50–100 $\mu m$, with little temporal variability. We therefore apply single-layer model scenarios with $r_e$ of 50 and 100 $\mu m$. We also apply the reported BC mixing ratio of 3 ppb.

- Casey et al. (2017): This study reports $\lambda = 0.36 - 2.5\,\mu m$ measurements of clean and heavily polluted snow from fossil fuel combustion near the South Pole during 2014–2015. We present comparisons with three of the albedo curves shown in Figure 7 of Casey et al. (2017) (which itself includes an independent comparison against an older version of SNICAR). Surface grain sizes were retrieved by comparing measured spectral albedo across the $1.03\,\mu m$ absorption feature with a lookup table of modeled reflectances from spherical ice grains (Nolin and Dozier, 2000). For consistency, we therefore also apply model spheres when using the reported grain sizes, though we include additional scenarios with hexagonal plates. We apply the reported BC and dust mixing ratios measured from each site, assuming the BC to be sulfate-coated and the dust to be fine Saharan dust. The reported dust concentrations are sufficiently low to have little impact on our analysis.

- Hadley and Kirchstetter (2012): Controlled laboratory experiments were performed in this study with varying amounts of flame-generated soot incorporated into aqueous suspensions that became mixed with quasi-spherical ice grains. We thus apply spherical ice grains in our model comparison with this dataset. Spectral albedo measurements, limited to $\lambda = 400 - 900\,nm$, were made on 5 cm thick snow samples with direct zenith illumination, and scaling factors were applied by the authors to derive semi-infinite snow albedo. BC mass mixing ratios up to 1680 ppb were generated in snow with multiple ice effective grain sizes, and we present the measured spectra shown in Figure 1 of Hadley and Kirchstetter (2012). The $k_a$ of the soot was estimated to be $15\,m^2\,g^{-1}$ at $\lambda = 532\,nm$, higher than our standard soot, so we apply our more absorptive sulfate-coated soot for this comparison, with implications discussed below.

- Brandt et al. (2011): This study also created artificial snow infused with commercial soot, but did so outdoors on a 75 $m^2$ field. The frozen water droplets were quasi-spherical with an estimated effective grain size of 60 $\mu m$. Spectral albedo measurements over $\lambda = 0.35 - 2.5\,\mu m$ were conducted under diffuse (cloudy) illumination on snow with and without a substantial load ($\sim 2500$ ppb) of BC. The "Aquablack-162" soot applied in this study had a smaller and narrower size distribution than the standard BC used in SNICAR, so for this comparison we created a separate species of BC with $r_m = 65\,nm$, $\sigma_g = 1.3$, and $k_a = 6.0\,m^2\,g^{-1}$ at $\lambda = 550\,nm$, matching the reported specifications by Brandt et al. (2011). (We also include this version of BC in the optics library.) A second artificial snowpack with a different type of soot was presented in this study but was not rigorously characterized and showed noisier albedo measurements, so we do not study it here.





– Aoki et al. (2000): These authors measured spectral albedo of natural snowpacks in Hokkaido, Japan, coincident with vertical profiles of snow grain size, density, and impurity concentrations. The bulk impurity concentrations were not partitioned by species, though mean particle absorptivity was derived. Mixing our standard BC and fine Saharan dust optical properties to match this absorptivity suggests a mean BC percentage by mass of $\sim 0.5 - 1.5\%$, though with considerable uncertainty. We assume a $98.5\%/1.5\%$ partitioning of dust and coated BC in our default scenarios for

comparison, but find in sensitivity studies that larger BC fractions improve the spectral agreement. We apply the 3-layer vertical profiles of snow grain size, density, and impurity loads shown in Figure 10 ("Model 4") and Figures 12a and 12c of Aoki et al. (2000) for our three comparisons. Albedo measurements span $\lambda = 0.35 - 2.4\,\mu m$, with a gap between $\sim 1.83$ and $1.91\,\mu m$.

    – Skiles and Painter (2017): Field measurements of spectral snow albedo ($0.35 < \lambda < 1.5\,\mu m$) and vertical profiles of snow

grain size, dust, and black carbon concentrations were conducted in this study throughout the 2013 spring melt season at the Senator Beck study site in the San Juan Mountains of Colorado, an area that periodically experiences large dust deposition events. We select three profiles for comparison: One with relatively clean snow before much melting had occurred (13 April), one with moderate dust loads after a deposition event (28 April), and one with very heavy surface dust burdens after an additional deposition event and substantial melt had occurred (3 May). The vertical profiles of

impurities, snow grain size, and density imposed in the model included measurements in 10 layers at $3\,cm$ resolution and an 11th deep layer. Snow grain size measured with contact spectroscopy represents a sphere-equivalent grain size, and thus we apply spheres for this comparison. We apply the Colorado dust optical properties and particle size distribution unique to this environment (Section 3.3.3, Skiles et al., 2017), along with the direct or diffuse light conditions that coincided with the measurements. For the 13 and 28 April cases we show a second model simulation with a thin surface

layer composed of smaller or larger snow grains to demonstrate that improved comparison is achievable. Furthermore, top-layer dust concentration is set to that of the second-from-top, dustier layer in this sensitivity simulation for 28 April.

## 5.1   Clean snow

Fig. 9 shows model–measurement comparisons of snow with extremely low LAC content. The comparison with Grenfell et al. (1994) also includes 2-layer models of spheres and hexagonal plates with a very thin $0.25\,mm$ surface layer. Because measured

grain size was not vertically-resolved to better than $5\,mm$, Grenfell et al. (1994) demonstrated how an unresolved but plausible thin surface layer of fine-grained snow could substantially improve model-measurement agreement in the NIR. Our model sphere scenario is identical to that utilized by Grenfell et al. (1994) and is included for reference. Fig. 9a shows that even better agreement at $\lambda > 2.0\,\mu m$ can be achieved with a hexagonal plate scenario. All scenarios slightly underpredict a single measurement at $1.9\,\mu m$. All scenarios also produce slightly higher albedo than measured at $\lambda < 0.4\,\mu m$.

The model comparison with Hudson et al. (2006) (Fig. 9b) applies hexagonal plates with grain size not well-constrained by independent measurements. In addition to the single-layer semi-infinite cases with $r_e$ of 50 and $100\,\mu m$, we also include a 3-layer model with $r_e = 75\,\mu m$ (top $0.25\,mm$), $r_e = 100\,\mu m$ ($0.25 - 0.50\,mm$ depth), and $r_e = 175\,\mu m$ below. This scenario





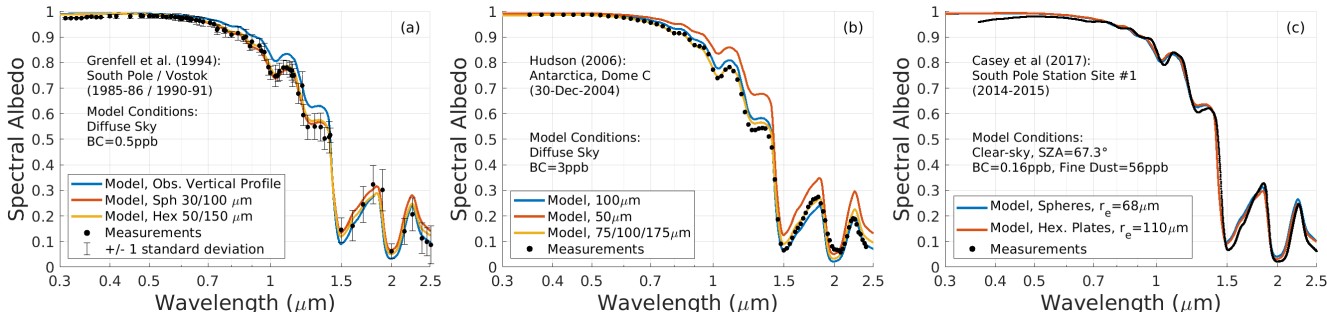

**Figure 9.** Comparison of modeled spectral snow albedo with measurements from Grenfell et al. (1994) (a), Hudson et al. (2006) (b), and Casey et al. (2017) (c) of very clean snow from Antarctica . The environmental conditions and model parameters for each setting are listed on the figures and described in the text.

was chosen again to illustrate that excellent agreement can be achieved across the NIR when very thin model layers are applied near the surface. Measurements from Hudson et al. (2006) at $\lambda < 0.4$ are slightly higher than those from Grenfell et al. (1994), resulting in better model-measurement agreement in the UV. Model curves are slightly too low at $\lambda = 1.9 - 2.0\,\mu m$ and slightly high at $\lambda = 1.2 - 1.35\,\mu m$, though the latter could probably be remedied with a 4- or 5-layer model.

The clean-snow case from Casey et al. (2017), depicted in Fig. 9c, shows strong agreement with the matching spherical-grain scenario from SNICAR-ADv3 and a hexagonal plate scenario with slightly larger effective grain size than the sphere $r_e$ retrieved by Casey et al. (2017). The agreement, however, is imperfect in the visible and UV spectrum, where the measurements show lower albedo than the model. This discrepancy may result from uncertainty in the measurements, which were of directional reflectance and required an anisotropic correction factor to determine albedo. We expect truly pristine snow to have higher visible and UV albedo, akin to what was measured by Hudson et al. (2006), unless the blue/UV ice absorption coefficients applied from Picard et al. (2016) and Warren and Brandt (2008) are biased low.

## 5.2 Snow with Black Carbon as the Dominant Source of LAC

Fig. 10 presents model-measurement comparisons from three studies of snow contaminated primarily with black carbon. The comparisons with Casey et al. (2017) (Fig. 10a–c), which are of natural snow with BC mass mixing ratios of 2400, 3300, and 490 ppb, are generally excellent. The spectral shape and magnitude of modeled albedo in the perturbed part of the spectrum ($\lambda < 1.0\,\mu m$) are near-perfect in panels (b) and (c), but do not perfectly match the measured albedo in Fig. 10a, which shows snow with a slightly more brownish hue than that produced by the standard SNICAR BC. Smaller size distributions of BC can be more absorptive in the blue than red (as with the Aquablack described below), creating a spectral response of the nature shown, but BC size distributions were not reported. The hexagonal plate model curves in Figs. 10b and 10c include a very thin (0.25 mm) surface layer with grain size selected to optimize agreement. The single-layer sphere cases are identical to those specified by Casey et al. (2017).





**Figure 10.** Comparison of modeled spectral snow albedo with measurements from Casey et al. (2017) (a–c), Hadley and Kirchstetter (2012) (d–f), and (Brandt et al., 2011) (g–h) of snow contaminated primarily with black carbon. The environmental conditions and model parameters for each setting are listed on the figures and described in the text.

The comparison against Hadley and Kirchstetter (2012), for snow $r_e$ of 55, 65, and $110\,\mu\mathrm{m}$ is shown in Fig. 10d–f. In general, this comparison is also quite promising throughout the visible spectrum and for the entire range ($0-1680\,\mathrm{ppb}$) of BC mixing ratios explored. We note, however, that there is a discrepancy between the estimated $k_a$ of the soot used in this study ($15\,\mathrm{m^2\,g^{-1}}$ at $\lambda = 532\,\mathrm{nm}$) and that of the coated BC applied in SNICAR ($11.5\,\mathrm{m^2\,g^{-1}}$). If a form of BC with $k_a = 15\,\mathrm{m^2\,g^{-1}}$ were applied in SNICAR, the model would overestimate the BC impacts on snow albedo, with albedo reductions roughly 30% greater than shown. A key finding from the measurements of Hadley and Kirchstetter (2012) is that the albedo perturbation from BC increases with increasing snow grain size, to a degree that is in-line with the model results presented here and confirming earlier theoretical arguments for this phenomenon (Warren and Wiscombe, 1980).





Fig. 10g compares SNICAR-ADv3 with measurements of clean and contaminated artificial snowpacks from Brandt et al. (2011), using the measured BC concentrations from filter samples and the measured snow grain size, along with our standard BC optical properties. The measurements are quite noisy near $\lambda = 1.4\,\mu\text{m}$ and, despite near-identical grain sizes of the two snowpacks, the albedos of clean and dirty snow diverge at longer wavelengths so we focus the comparison on shorter wavelengths. Agreement is reasonable in this case, but Fig. 10h shows that improved agreement results when more detailed knowledge of the impurity properties is incorporated. First, Brandt et al. (2011) note that the city water supply used to create the snow "contained a small amount of a brown absorber (perhaps rust or humic acid)". Hence instead of applying $12\,\text{ppb}$ of BC for the clean snow case, we apply $100\,\text{ppb}$ of brown carbon, which produces an improved spectral match with the clean snow case. Second, when we instead apply the Aquablack BC properties described above, the modeled albedo of the contaminated snow takes on a slightly more brownish hue, leading to a near-perfect match with measurements. Finally, Brandt et al. (2011) speculated that the hydrophillic soot they used may have resided inside of the ice grains, though they could not confirm this with observations and found that modeling with an external mixture of BC, as done here, produced good albedo agreement. Internally-mixed BC in the model would produce a larger albedo impact (He et al., 2017), worsening the albedo comparison. The snow produced by Hadley and Kirchstetter (2012) may also have contained internally-mixed BC because the soot was incorporated into an aqueous suspension, but this is also unconfirmed.

### 5.3 Snow with Dust and Black Carbon

BC and dust constitute the dominant sources of LAC in most natural snowpack, though their relative proportions vary widely in space and time. Measurements by Aoki et al. (2000) of natural snow with moderate impurity loads are shown in Fig. 11a–c. When we combine a spherical grain model, as used by Aoki et al. (2000), with the same $r_e$ profiles they applied, we find similarly excellent agreement with measurements at $\lambda > 1.5\mu\text{m}$ and slight model overprediction at shorter NIR wavelengths, similar to the comparison they presented with a different radiative transfer model. Assuming hexagonal plates with larger grain sizes, which are encouragingly more similar to the $r_e$ values derived through image processing by Aoki et al. (2000), also produces good agreement in the NIR, especially near the $1.3\,\mu\text{m}$ ice absorption feature. Adding a $0.25\,\text{mm}$ fine-grain layer produces excellent agreement throughout the NIR, depicted with the yellow curve in Fig. 11a. (For economy, we omit this hypothetical layer in panels (b) and (c), though it also reduces model bias in those comparisons). Using the loosely-constrained BC/impurity mass fraction of 1.5% produces an excessive visible albedo for the scenes shown in Figs. 11a and 11b, but reasonable albedo for that shown in Fig. 11c. The scene in Fig. 11b is too brown and too dark to be replicated with our standard coated BC and dust, constrained with measured impurity loads, but by assigning 15% of the impurity mass to brown carbon we achieve an excellent match. This is merely illustrative, as Aoki et al. (2000) do not mention brown carbon, though its presence can't be ruled out either. It is also quite possible that more absorptive (brown) dust was present than represented in our model. The observation of $\sim 20\times$ higher impurity loads at the snow surface than at depth, as exhibited in Figs. 11a and 11c, demonstrates the importance of accurately resolving the vertical distribution of impurity load, especially after melt consolidation has occurred (Doherty et al., 2013).



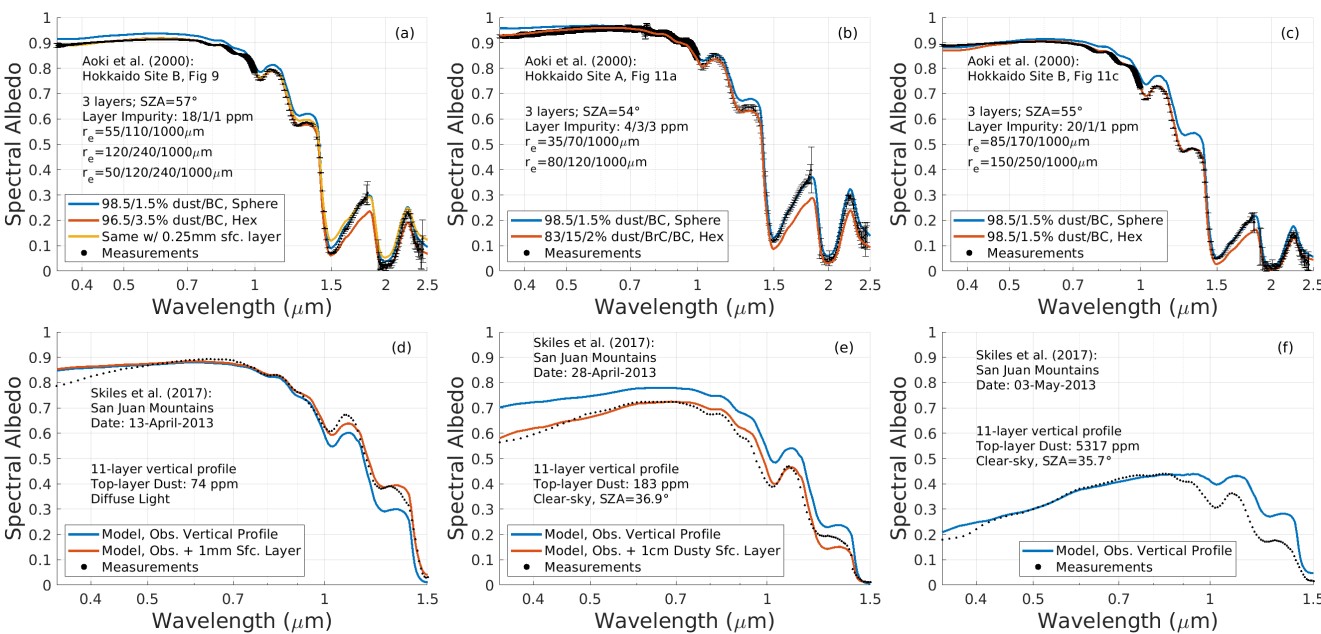

**Figure 11.** Comparison of modeled spectral snow albedo with measurements from Aoki et al. (2000) (a–c) and Skiles and Painter (2017) (d–f) of snow contaminated with black carbon and dust. The environmental conditions and model parameters for each setting are listed on the figures and described in the text. The sets of layered effective snow grain sizes ($r_e$) for model runs applied in comparison with Aoki et al. (2000) are listed on panels a–c. Model comparisons with Skiles and Painter (2017) apply their multi-layer measurements of dust, black carbon, and snow grain size at $\sim 3\,\mathrm{cm}$ vertical resolution

The snow measured by Skiles and Painter (2017) was contaminated primarily by dust, in widely varying amounts, and secondarily by black carbon. Using the Colorado dust properties and measured vertical impurity distributions in our modeling produces good agreement in visible albedo for the 13-April case, insufficient darkening in the 28-April case, and excellent agreement in the extreme dust case of 3-May (Fig. 11d–f). The measured dust mixing ratio in the second-from-top snow layer for the 28-April case was three times larger than that of the top layer. A sensitivity study with a new $1\,\mathrm{cm}$ surface layer

darkened with this larger dust concentration and a larger snow grain size produces good agreement (red curve in Fig. 11e). This is merely illustrative, but possibly within the range of measurement uncertainty, considering the different footprints of spectral albedo and snow-pit excavations used for vertical profiles. Modeled albedo in the NIR is overpredicted in the two most dusty cases (28-April and 3-May). Skiles et al. (2017) also observed this phenomenon and discuss possible explanations including under-estimated snow grain size due to dust masking of the $1.03\,\mu\mathrm{m}$ ice absorption feature and complex interactions associated

with internal dust–ice mixtures that are not represented in SNICAR. Our demonstration of improved NIR agreement with grain size adjustments alone for the two April cases points to potential issues with measurements of grain size and shape, while our inability to resolve the NIR bias in the dusty 3-May case without wrecking the visible agreement suggests a need to further investigate the potential role of mixing state described by Skiles et al. (2017). Overall, however, we find the agreement between





measurements and modeling across this large range of dust-induced darkening, with visible albedo ranging from $0.31 - 0.86$,
to be encouraging.

## 6   Conclusions

We have presented the formulation of a publicly-available model and accompanying library of particle optical properties
that are used to simulate the spectral albedo of snow, dependent on many variables including the content of light-absorbing
constituents (LAC). Types of LAC included with SNICAR-ADv3 are black carbon, brown carbon, mineral dust, volcanic ash,
and snow algae, with variants associated with coatings, particle size, mineralogy (for dust), and pigment content (for algae).
SNICAR-ADv3 unifies numerous improvements and capabilities that have been introduced to the model through separate
studies, including application of the adding-doubling two-stream solver (Dang et al., 2019) and representations of non-spherical
ice particles (He et al., 2017), carbon dioxide snow (Singh and Flanner, 2016), snow algae (Cook et al., 2017), different types of
dust (Polashenski et al., 2015; Skiles et al., 2017; Wolff et al., 2009) and volcanic ash (Flanner et al., 2014), new ice refractive
indices (Picard et al., 2016), and SZA-dependent surface spectral irradiances. For representing natural snow with unknown ice
grain shape, we recommend applying one of the three available non-spherical ice grain shapes (spheroids, hexagonal plates,
or Koch snowflakes), which have smaller scattering asymmetry parameters than spheres and therefore reduce the radiative
penetration depth and exposure of sub-surface LAC (Dang et al., 2016). The use of hexagonal plates instead of equal-SSA
spheres reduces the simulated albedo impact of black carbon in single-layer simulations by $\sim 24\%$.
Compared with spectral albedo measurements of clean snow, SNICAR-ADv3 performs well across the solar spectrum, and
arguably better when non-spherical ice shapes are used, but in most cases a very thin ($\sim 0.25\,\mathrm{mm}$) surface layer composed
of fine-grain snow must be introduced to match observations across the near-infrared spectrum. No observational studies, to
our knowledge, have resolved snow grain size at this vertical resolution, so although the presence of such layers is physically
plausible, they are not confirmed to exist. Compared to spectral albedo measurements of snow laden with black carbon, dust,
and (in one case) brown carbon, SNICAR-ADv3 also performs well when reasonable assumptions are made. A substantial
source of uncertainty in some comparisons, however, is the optical properties of the LAC present in the snow being studied, as
these vary substantially with source provenance, particle size, and atmospheric processing. LAC internally-mixed in weakly-
absorbing media, including (e.g.,) sulfate and ice, are predicted to generally absorb $\sim 20 - 60\%$ more solar energy per unit
mass than externally mixed LAC, but the mixing states of particles in snow are also rarely known. We see no systematic
directional bias in the snow albedo reduction simulated by SNICAR-ADv3 when applying the standard LAC included in
the optical property library, suggesting that these properties are reasonable to apply when site-specific LAC properties are
unknown. A companion study (*Whicker et al.*, in prep.) extends SNICAR-ADv3 to represent glacial ice. Other target areas for
model improvement include representation of liquid water and snow roughness (Larue et al., 2020), as well as experimental
verification of simulated impacts from snow algae and volcanic ash. It is likely that algae modeling techniques will need to be
adapted. We hope that the web-based model, multi-layer source code, and accompanying particle optics library of SNICAR-
ADv3 prove useful for future research, educational, and model development efforts.





**Appendix A:  The delta-Eddington adding-doubling solution**

We refer readers to Briegleb and Light (2007) for a detailed derivation of the delta-Eddington adding-doubling multi-layer two-stream solution. Here, we present only the essential equations applied in SNICAR-ADv3.

Defining $f = g^2$, the layer bulk optical properties (Eqs. 4–6) are transformed with delta-scalings to account for forward scattering as:

$$\tau^* = (1 - \omega f)\tau \tag{A1}$$

$$\omega^* = \frac{(1-f)\omega}{1-\omega f} \tag{A2}$$

$$g^* = \frac{g-f}{1-f} \tag{A3}$$

The following intermediate variables are applied in the two-stream solution, with Eqs A7 and A8 dependent on the cosine of the incident zenith angle ($\mu_0$):

$$\Gamma = \sqrt{3(1-\omega^*)(1-\omega^* g^*)} \tag{A4}$$

$$u = \frac{3}{2}\left(\frac{1-\omega^* g^*}{\Gamma}\right) \tag{A5}$$

$$N = (u+1)^2 e^{\Gamma\tau^*} - (u-1)^2 e^{-\Gamma\tau^*} \tag{A6}$$

$$\eta = \frac{3}{4}\omega^*\mu_0\left(\frac{1+g^*(1-\omega^*)}{1-\Gamma^2\mu_0^2}\right) \tag{A7}$$

$$\gamma = \frac{1}{2}\omega^*\left(\frac{1+3g^*(1-\omega^*)\mu_0^2}{1-\Gamma^2\mu_0^2}\right) \tag{A8}$$

The layer reflectance to direct-beam radiation is:

$$R(\mu_0) = \frac{1}{N}(\eta+\gamma)(u^2-1)\left(e^{\Gamma\tau^*}-e^{-\Gamma\tau^*}\right) + (\eta-\gamma)\left[\frac{4u}{N}e^{\frac{-\tau^*}{\mu_0}}-1\right] \tag{A9}$$



The layer transmittance to direct radiation is:

$$T(\mu_0) = (\eta + \gamma)\frac{4u}{N} + \left[\frac{1}{N}(\eta - \gamma)(u^2 - 1)(e^{\Gamma\tau^*} - e^{-\Gamma\tau^*}) - \eta - \gamma + 1\right]e^{\frac{-\tau^*}{\mu_0}} \tag{A10}$$

The layer reflectance and transmittance to diffuse radiation are calculated by integrating the respective direct quantities over $\mu$:

$$\overline{R} = 2\int_0^{+1} \mu R(\mu)d\mu \tag{A11}$$

$$\overline{T} = 2\int_0^{+1} \mu T(\mu)d\mu \tag{A12}$$

where eight Gaussian angles are used for the numerical integration.

Properties of individual layers are then combined through the adding-doubling approach to produce bulk properties for multiple layers. The reflectance and transmittance of two stacked layers, with layer 1 overlying layer 2 and direct-beam illumination on layer 1 are:

$$R_{12}(\mu_0) = R_1(\mu_0) + \frac{\left[\left(T_1(\mu_0) - e^{-\tau_1^*/\mu_0}\right)\overline{R}_2 + e^{-\tau_1^*/\mu_0}R_2(\mu_0)\right]\overline{T}_1}{1 - \overline{R}_1\overline{R}_2} \tag{A13}$$

$$T_{12}(\mu_0) = e^{-\tau_1^*/\mu_0}T_2(\mu_0) + \frac{\left[\left(T_1(\mu_0) - e^{-\tau_1^*/\mu_0}\right) + e^{-\tau_1^*/\mu_0}R_2(\mu_0)\overline{R}_1\right]\overline{T}_2}{1 - \overline{R}_1\overline{R}_2} \tag{A14}$$

The bulk properties for these two combined layers under diffuse illumination of layer 1 are:

$$\overline{R}_{12} = \overline{R}_1 + \frac{\overline{T}_1\overline{R}_2\overline{T}_1}{1 - \overline{R}_1\overline{R}_2} \tag{A15}$$

$$\overline{T}_{12} = \frac{\overline{T}_1\overline{T}_2}{1 - \overline{R}_1\overline{R}_2} \tag{A16}$$

The combined transmittances for illumination from below are identical whereas the reflectance for illumination from below, i.e. by upward scattered radiation that is assumed to be diffuse, is equivalent to Eq. A15 but with reversed layer subscripts.

The cumulative layer interface properties for downward and upward propagating radiation are solved through loops that integrate from the top–down and bottom–up of the column. These interface quantities are:

– $\tau^*$: the scaled optical depth from the model top to the interface





– $R_{up}(\mu_0)$: the reflectance of the entire column below the interface to downwelling direct radiation from above

     – $\overline{R}_{up}$: the reflectance of the column below the interface to diffuse radiation from above

     – $\overline{R}_{dn}$: the reflectance of the column above the interface to upwelling diffuse radiation from below

     – $T_{dn}(\mu_0)$: the bulk transmittance between the model top and the interface with respect to direct radiation incident on the model top

– $\overline{T}_{dn}$: the transmittance between the model top and the interface to diffuse radiation

With these quantities defined, the downwelling and upwelling fluxes at each layer interface ($F^{\downarrow}_{dr}$, $F^{\uparrow}_{dr}$), normalized to the unit direct flux incident at the model top, are, respectively:

$$F^{\downarrow}_{dr} = e^{-\tau^*/\mu_0} + \frac{(T_{dn}(\mu_0) - e^{-\tau^*/\mu_0}) + e^{-\tau^*/\mu_0} R_{up}(\mu_0)\overline{R}_{dn}}{1 - \overline{R}_{dn}\overline{R}_{up}} \tag{A17}$$


$$F^{\uparrow}_{dr} = \frac{e^{-\tau^*/\mu_0} R_{up}(\mu_0) + (T_{dn}(\mu_0) - e^{-\tau^*/\mu_0}) + \overline{R}_{up}}{1 - \overline{R}_{dn}\overline{R}_{up}} \tag{A18}$$

The downwelling and upwelling fluxes at each layer interface ($F^{\downarrow}_{df}$, $F^{\uparrow}_{df}$), normalized to the unit diffuse flux incident at the model top, are, respectively:

$$F^{\downarrow}_{df} = \frac{\overline{T}_{dn}}{1 - \overline{R}_{dn}\overline{R}_{up}} \tag{A19}$$

$$F^{\uparrow}_{df} = \frac{\overline{T}_{dn}\overline{R}_{up}}{1 - \overline{R}_{dn}\overline{R}_{up}} \tag{A20}$$

Finally, the albedo is derived from fluxes at the top interface:

$$\alpha = \frac{F^{\uparrow}_{dr}(top) + F^{\uparrow}_{df}(top)}{F^{\downarrow}_{dr}(top) + F^{\downarrow}_{df}(top)} \tag{A21}$$



*Code and data availability.* The web-based single-layer version of SNICAR-ADv3 can be run at: http://snow.engin.umich.edu. Matlab source code for the multi-layer model, and the accompanying netCDF library of particle optics files are available at: https://github.com/mflanner/SNICARv3. A frozen branch of the code used in this manuscript is associated with the doi: 10.5281/zenodo.5176213. The scripts and data used to generate all plots are available at: http://snow.engin.umich.edu/snicarv3_analysis/.

*Author contributions.* MGF wrote the manuscript, assimilated the code updates, and created the web-based model. JA performed model
evaluations against spectral albedo measurements, JMC created the original representation of snow algae, CD and CSZ incorporated the adding-doubling solver, CH designed and incorporated the representation of non-spherical ice particles, XH provided top-of-atmosphere spectral irradiances, DS incorporated carbon dioxide snow and Martian dust optical properties, SMS provided Colorado dust optical properties, CAW merged the adding-doubling solver into our new code base, and CSZ and MGF created the original SNICAR model. All authors reviewed the manuscript.

*Competing interests.* The authors declare no competing interests.

*Acknowledgements.* We thank Teruo Aoki, Kimberly Casey, Odelle Hadley, and Richard Brandt for providing us with spectral albedo measurements from their studies. This work was supported by grant OPP-1712695 from the U.S. National Science Foundation.





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
