# Peer review of "SNICAR-AD v3: A Community Tool for Modeling Spectral Snow Albedo"

_Geoscientific Model Development, 2021_

## Referee Comment (RC1)

**Review of « SNICAR-AD V3: A community Tool for Modeling Spectral Snow Albedo»**
**by Flanner et al.** (gmd-2021-182)

*Marie Dumont*

**Summary and recommendations**

This paper presents a new release of the snow spectral albedo model, SNICAR-AD v3 including the most recent developments in terms of radiative transfer solver, ice refractive index and light absorbing particles. It was a great pleasure to review this paper. This is a very important and useful study and release considering the wide range of users of SNICAR and the relevance of the model. I think the model is extremely beneficial for the climate and the cryosphere community and beyond (extra-terrestrial studies). I only have a few general and minor comments that I am detailing below.

**General comments.**

1 – In the model evaluation (section 5) and the conclusions, the need for a very thin layer of snow with high SSA is underlined to better reconcile the measurements and the simulations in the NIR. The agreement in the NIR wavelengths can also be improved by using alternative values of the ice refractive index in these spectral ranges (e.g. Carmagnola et al., 2013 fig. 13 et 14 – with spheres, Dumont et al., 2021 with the theoretical of Malinka, 2014; fig. 10). I was wondering if this has been tested as an alternative to the very thin layer in SNICAR-AD.

2 – Still in the model evaluation (section 5), most of the comparison between the model and the measurements are visual and qualitative. I was wondering if some more quantitative information (e.g. broadband albedo bias, RMSE …. maybe just at least order of magnitude ?) can be added to ease the comparison with other models and other studies ?

3 – Many different acronyms are used in the literature for light absorbing particles (constituents, impurities, … ), LAP, LAC, LAI. In this paper, LAC and light absorbing particles are used. Is there a reason for using both ?

**Minor comments.**

P2 line 21 - "the morphology and size of the ice grains" . The use of the term "ice grains" is a bit ambiguous since snow can be seen a porous media, and it's not always easy to segment "grains". Grains might also refer to crystallographic orientation (e.g. Montagnat et al., 2020). "ice grains" is however useful for defining re. I tend to refer to the morphology of the snow microstructure instead.

P2 line 40 – TARTES uses the Kokhanovsky and Zege formalism only for the single scattering properties.

P4 line 93 – Maybe the definition the extinction optical thickness could be helpful

P4 line 94 – Is it "single-scatter" or "single-scattering" albedo ?

P4 line 96 – Maybe details about the meaning of g =0, g=1 could be also helpful

Section 2.2 (p5). Why was the resolution of 10 nm chosen ? I guess it's a compromise between the numerical efficiency and the accuracy ?  Is the ice refractive index averaged over a 10 nm spectral bands ? Or is the value of the central wavelengths used ?

Section 2.3 Maybe it could be useful to shortly describe here the underlying hypothesis in terms of optics (independent scatterers ..) that allows the writing of Eq. 4-6.

P6 Eq 7. What is the spectral resolution used to compute F(lambda) ?

P7 Table 1 – What are the values of ozone used ?

P7 line 177 – I agree that the spectral irradiance and the broadband albedo are sensitive to the RT model, and TOA irradiance. They also greatly depends on the atmospheric profiles and cloud properties ...

P7 line 183 – What are the types of aerosols used ?

P 9 Eq 9 (and at several places in the paper) – What is the reason for the choice of a log normal distribution ?

P10 line 232 – Maybe a very short sentence to justify the selection of the 4 shapes could be helpful (maybe it's already somewhere and I missed it).

P12 line 294 – "MAC", I guess it's "ka". Maybe the acronym should be added line 289.

Figure 3 c,e, just out of personal curiosity from what are the little bump around 400 nm for fine dust coming ?

Figure 7 – I think "(FGBC)" is missing in the caption.

Section 5 – For the clear sky measurements, where the direct/diffuse irradiance ratio used in the model compared to the one measured in the field (if any) ? This could also impact the comparison.

P4  line 793 - "less exposure of sub-surface LAC", the reason is then the same of the increase effect of LAC for "large grains" (p 31 lines 729-731) ?

*Code availability*
The code, the web app and the library are well structured, documented and easily accessible. I was wondering if it is possible to add a *readme* file for the LAC properties. The .nc are self-documented but maybe it could help the reader to know which properties are required to implemented a new type of LAP ?
The code is in matlab. Is there any plan to have an 'open language' version ?
For the web-app is it possible to use "ground" albedo that would be not constant with wavelength ?

**References used in the review.**

- Carmagnola, C. M., Domine, F., Dumont, M., Wright, P., Strellis, B., Bergin, M., Dibb, J., Picard, G., Libois, Q., Arnaud, L., and Morin, S.: Snow spectral albedo at Summit, Greenland: measurements and numerical simulations based on physical and chemical properties of the snowpack, The Cryosphere, 7, 1139–1160, https://doi.org/10.5194/tc-7-1139-2013, 2013.

- Dumont, M., Flin, F., Malinka, A., Brissaud, O., Hagenmuller, P., Lapalus, P., Lesaffre, B., Dufour, A., Calonne, N., Rolland du Roscoat, S., and Ando, E.: Experimental and model-based investigation of the links between snow bidirectional reflectance and snow microstructure, The Cryosphere, 15, 3921–3948, https://doi.org/10.5194/tc-15-3921-2021, 2021.

- Malinka, A. V. (2014). Light scattering in porous materials: Geometrical optics and stereological approach. Journal of Quantitative Spectroscopy and Radiative Transfer, 141, 14-23.

- Montagnat M, Löwe H, Calonne N, Schneebeli M, Matzl M and Jaggi M (2020) On the Birth of Structural and Crystallographic Fabric Signals in Polar Snow: A Case Study From the EastGRIP Snowpack. Front. Earth Sci. 8:365. doi: 10.3389/feart.2020.00365

---

## Author Comment (AC1)

We thank the reviewers, Marie Dumont and Ghislain Picard, for providing helpful comments. Below, we respond to these comments and identify associated changes to our manuscript that we have made. Referee comments are in black and our responses are in blue.

**Review #1 (Marie Dumont):**

**Summary and recommendations**

This paper presents a new release of the snow spectral albedo model, SNICAR-AD v3 including the most recent developments in terms of radiative transfer solver, ice refractive index and light absorbing particles. It was a great pleasure to review this paper. This is a very important and useful study and release considering the wide range of users of SNICAR and the relevance of the model. I think the model is extremely beneficial for the climate and the cryosphere community and beyond (extra-terrestrial studies). I only have a few general and minor comments that I am detailing below.

**General comments.**

1 – In the model evaluation (section 5) and the conclusions, the need for a very thin layer of snow with high SSA is underlined to better reconcile the measurements and the simulations in the NIR. The agreement in the NIR wavelengths can also be improved by using alternative values of the ice refractive index in these spectral ranges (e.g. Carmagnola et al., 2013 fig. 13 et 14 – with spheres, Dumont et al., 2021 with the theoretical of Malinka, 2014; fig. 10). I was wondering if this has been tested as an alternative to the very thin layer in SNICAR-AD.

This is an excellent point, and we appreciate the reminder and notification of these studies. We have added text to section 5.1 and the Conclusions of our manuscript citing these studies and identifying uncertainty in NIR ice refractive index data as an alternative explanation for the bias in modeled albedo between wavelengths of 1.4 and 1.9 µm. We agree with the conclusion of *Dumont et al.* (2021) that additional careful measurements of ice complex refractive indices in the NIR are needed, and now include a similar point in our Conclusions section.

2 – Still in the model evaluation (section 5), most of the comparison between the model and the measurements are visual and qualitative. I was wondering if some more quantitative information (e.g. broadband albedo bias, RMSE .... maybe just at least order of magnitude ?) can be added to ease the comparison with other models and other studies ?

We appreciate the suggestion. We computed the RMSE for all model-measurement comparisons shown in Figures 9-11 and provide these as a spreadsheet in the supplementary material of analysis scripts and data (specifically, at: http://snow.engin.umich.edu/snicarv3 analysis/data/RMSE.ods). We chose not to include this in the main text, however, because we feel it detracts from what is already a long (~50 page) description of the model, and we question the utility of these higher-order statistics for several reasons. Our focus in this section is on the spectral differences and similarities between model and measurements, and much of this comparison becomes lost with higher-order statistics like broadband albedo and RMSE. Comparison of such statistics between studies also becomes problematic because (1) different sets of measurements have different spectral ranges, and thus broadband albedo has different weights between studies and RMSE is averaged over different domains (for example, the *Hadley and Kirchetetter* (2012) measurements only extend to 0.9 µm); (2) the downwelling spectral irradiances (weights for broadband albedo) are unique to each set of measurements yet are unavailable from most studies; and (3) noise and missing data in the measurements complicate the interpretation of statistics like RMSE (for example, RMSE of the model curves with high BC loads in Figs. 10g and 10h are both large and nearly identical (0.104 and 0.103, respectively) because of noise in the measurements from *Brandt et al* (2011) near 1.4 µm, and yet the model comparison shown in Fig. 10h is clearly superior to that shown in Fig. 10g. We could finesse the statistics to account for these factors, but we prefer to leave spectral comparisons to our descriptions and the eyes of the readers.

3 – Many different acronyms are used in the literature for light absorbing particles (constituents, impurities, ...), LAP, LAC, LAI. In this paper, LAC and light absorbing particles are used. Is there a reason for using both ?

We struggled to identify a universal term for the non-ice species in snow. Although the community has settled somewhat on using "light absorbing particles" (after moving on from "light absorbing impurities"), the present study also deals with algae, and we do not think it is accurate or appropriate to refer to algae as a "particle". We therefore settled on "light absorbing constituents (LAC)" as an all-encompassing term for particles and algae, while recognizing that LAP continues to be a useful term for the non-cellular/non-living species like dust and black carbon. Our use of "light absorbing particles" was accidental, and we have changed all such references to LAC. We continue to use the word "particle" in many instances where appropriate (e.g., particle size, particle shape, etc). We advocate for the adoption of "LAC" by others, especially when referring to situations that might include algae or other cellular matter.

Minor comments.

P2 line 21 - "the morphology and size of the ice grains". The use of the term "ice grains" is a bit ambiguous since snow can be seen a porous media, and it's not always easy to segment "grains". Grains might also refer to crystallographic orientation (e.g. Montagnat et al., 2020). "ice grains" is however useful for defining re. I tend to refer to the morphology of the snow microstructure instead.

We agree, and because this is in the introductory material, we changed this passage simply to: "ice microstructure".

P2 line 40 – TARTES uses the Kokhanovsky and Zege formalism only for the single scattering properties.

Changed "...formulations from..." to "... formulations of single scatter properties from ..."

P4 line 93 – Maybe the definition the extinction optical thickness could be helpful

This is defined in Equations 1 and 4, so we simply added "(See Eqs. 1 and 4)".

P4 line 94 – Is it "single-scatter" or "single-scattering" albedo ?

In fact, both are widely used in the literature, so it is not clear which is better to use. Several radiative transfer textbooks (e.g., *Petty*) and classic journal articles (e.g., *Wiscombe and Warren*, 1980) use "single scatter albedo", as well as the American Meteorology Society (AMS) Glossary of Meteorology, and hence we choose to continue using this term.

P4 line 96 – Maybe details about the meaning of g =0, g=1 could be also helpful

We agree. We added "Values of -1 and 1 imply perfect back and forward scattering, respectively."

Section 2.2 (p5). Why was the resolution of 10 nm chosen ? I guess it's a compromise between the numerical efficiency and the accuracy ? Is the ice refractive index averaged over a 10 nm spectral bands ? Or is the value of the central wavelengths used ?

The choice of 10 nm spacing was indeed a compromise between numerical efficiency and accuracy. All Mie optical property calculations were performed using native refractive index data that were linearly interpolated to the SNICAR spectral grid (i.e., to the band center wavelengths). Original ice real and imaginary refractive index data used by *Warren and Brandt* (2008) were interpolated to wavelengths in log-linear and log-log space, respectively (by those authors). The average wavelength spacing of data provided by *Warren and Brandt* (2008) over 0.2-5.0µm is 20nm and 5nm, respectively, so interpolation of these data to the 10nm SNICAR grid is tightly constrained regardless of the interpolation scheme.

Section 2.3 Maybe it could be useful to shortly describe here the underlying hypothesis in terms of optics (independent scatterers ..) that allows the writing of Eq. 4-6.

Yes, thanks for the reminder. We inserted the following text just before the introduction of these equations: "The bulk layer optical properties are then formulated using the traditional technique that assumes particles scatter independently in one another's far fields, which is generally valid when inter-particle spacing is substantially longer than the wavelength of interacting light. This assumption is evaluated in detail by *Wiscombe and Warren* (1980, Section 7), who concluded that although the assumption is not strictly valid for snow, its contribution to bias in snow albedo modeling is likely quite small and with unclear sign, at least when snow density is less than  $\sim 450 \text{ kg m}^{-3}$ ."

P6 Eq 7. What is the spectral resolution used to compute F(lambda)?

We added the following text shortly after this equation: "To obtain *f*t, we interpolate the output of SWNB2, whose native spectral resolution ranges from 0.3 to 25 nm, to the SNICAR spectral grid.

P7 Table 1 – What are the values of ozone used ?

We added the ozone burdens (in Dobson units) to this table.

P7 line 177 – I agree that the spectral irradiance and the broadband albedo are sensitive to the RT model, and TOA irradiance. They also greatly depends on the atmospheric profiles and cloud properties ...

Indeed! We reformulated this sentence to read: "Surface spectral irradiances and [broadband albedo] are sensitive to cloud and atmospheric conditions (e.g., *Wiscombe and Warren*, 1980), and can also be somewhat sensitive to the radiative transfer model and top-of-atmosphere irradiance data used (*Bair et al.*, 2019)."

P7 line 183 – What are the types of aerosols used ?

The SWNB model can incorporate any number of aerosol types, but the aerosol properties assumed for these fixed-AOD calculations are those of sulfate. The AOD is only 0.05, so the type of aerosol does not strongly impact these calculations. We added "with properties representative of sulfate" to the passage in question.

P 9 Eq 9 (and at several places in the paper) – What is the reason for the choice of a log normal distribution ?

There is a substantial history of assuming log-normal size distributions for snow grains (e.g., *Colbeck*, 1987, doi: 10.1016/0001-6160(87)90105-2) and aerosols, based partially on measurements showing common distributions of this form and also because they are amenable to analytical mathematical treatment. Some studies of aerosols alternatively apply gamma, multi-modal, and other size distributions, depending on the application and system being represented. For our purposes, however, the form of the distribution is not that important, as long as it is wide enough to average out the Mie resonance features. Especially for snow, different size distributions with the same effective radius (or specific surface area) will exhibit very similar bulk optical properties, regardless of the shape of the distribution, and hence other distributions could likely be used with similar results.

P10 line 232 – Maybe a very short sentence to justify the selection of the 4 shapes could be helpful (maybe it's already somewhere and I missed it).

We expanded this sentence so it now reads: "Finally, users can specify one of four ice particle shapes, identified by *He et al.* (2017) as a sample of representative shapes that produce diverse snow optical properties".

P12 line 294 – "MAC", I guess it's "ka". Maybe the acronym should be added line 289.

Thanks for catching this. Since this was our only use of "MAC" in the paper, we simply changed it to k\_a.

Figure 3 c,e, just out of personal curiosity from what are the little bump around 400 nm for fine dust coming ?

This is a feature of the hematite refractive indices (http://eodg.atm.ox.ac.uk/ARIA/) we used to model these two types of dust. Below is a figure of the imaginary component of the hematite refractive index, showing strong absorption in the blue part of the spectrum. The bump appears more prominently in Mie results with small particle sizes (and obviously also with larger hematite volume fractions).

Figure 7 – I think "(FGBC)" is missing in the caption.

Thanks. We added this to the caption.

Section 5 – For the clear sky measurements, where the direct/diffuse irradiance ratio used in the model compared to the one measured in the field (if any) ? This could also impact the comparison.

This is a good point, and one reason why diffuse-sky albedo measurements can be preferable for model evaluation. We added the following to Section 5.1: "Another source of uncertainty when evaluating albedo under clear-sky conditions is the fraction of diffuse light, especially at short wavelengths, that was present for the measurements. Generally lacking this information, we assume unidirectional irradiance for comparisons against the clear-sky cases."

P4 line 793 - "less exposure of sub-surface LAC", the reason is then the same of the increase effect of LAC for "large grains" (p 31 lines 729-731) ?

Yes, exactly. The scattering asymmetry parameter (determined both by grain size and shape) governs the light penetration depth and exposure of sub-surface LAC.

Code availability

The code, the web app and the library are well structured, documented and easily accessible. I was wondering if it is possible to add a readme file for the LAC properties. The .nc are self-documented but maybe it could help the reader to know which properties are required to implemented a new type of LAP ?

In fact, a key motivation for preparing this manuscript was to provide detailed information about the LAC properties, underlying assumptions that were adopted, and how specifically they were produced. We believe the paper, combined with the metadata included in the netCDF files of LAC optical properties, provides an adequate description of the properties and key information needed for reproducibility and/or creation of new properties. It is unclear to us what information is being requested for a readme file that is not already in the manuscript or metadata, but if this is clarified we would be happy to provide one.

The code is in matlab. Is there any plan to have an 'open language' version ?

Yes. Co-author Joseph Cook has developed a Python implementation of the model which includes capabilities both of SNICAR-ADv3 and modifications for representing ice albedo described by *Whicker et al.* (2021, doi: 10.5194/tc-2021-272). This Python implementation is available at https://github.com/jmcook1186/BioSNICAR GO PY, and we now include this link in the manuscript.

For the web-app is it possible to use "ground" albedo that would be not constant with wavelength ?

We appreciate this suggestion and will explore such an option for future releases of the model. Possible ways to implement this could be: (1) to allow users to upload a file of spectrally-varying ground albedo (though this could be rather cumbersome), (2) have users define albedo in a few parts of the spectrum (e.g., blue, green, red, and near-IR), and/or (3) present a menu of surface types, each of which has a predefined surface spectral albedo. Option #3 is used for the online Snow TARTES model and seems like an attractive solution.

**Review #2 (Ghislain Picard):**

The paper entitled "SNICAR-AD v3: A Community Tool for Modeling Spectral SnowAlbedo" describes a new version of the well known SNICAR model to compute snow albedo. This article is motivated by the numerous scattered updates to the model performed since the initial release a decade ago.

The paper provides a very clear and comprehensive description of the model with numerous references on all the inputs used to parametrize the model. This is an impressive encyclopedic work that will benefit to the future users. The results section illustrates model capabilities in a few selected situations but does not provide a complete validation and does not highlight the challenges to conduct the simulations. Whilst I expect that most of the error comes from the input parametres uncertainties in most common cases, it would have been interesting to show the model intrinsic accuracy in the corner cases, for instance when using a 0.25mm thick layers or when using the model with 89° solar zenith angle.

The paper is in a perfect form, I recommend the paper to be accepted as it.

We appreciate these remarks. We also hope that this manuscript will serve as a useful reference for future users. There is indeed more work to be done on model evaluation. We agree with your suggestion to focus on model behavior in limiting cases, especially for thin snow, where potential exists for well-controlled observational studies and comparison with 3-D radiative transfer models.

A few points:

L80: "http://snowtartes.pythonanywhere.com/" has moved to http://snow.univ-grenoble-alpes.fr/snowtartes

Changed.

L82: "http://snowslope.pythonanywhere.com/" has moved to http://snow.univ-grenoble-alpes.fr/snowslope

Changed.

L287: "one size fits all". Better to avoid this (any) idiomatic expression.

Yes, good point. We removed this cliché phrase.

L 372: add a reference for the Bruggeman mixing approximation.

Added the original *Bruggeman* (1935) reference here.

L469: I don't understand "shape-preserving extrapolating functions".

This is a term used in Matlab documentation to describe the piecewise cubic function we employed, but we realize this is an odd description, so we changed it to simply "extrapolating functions", which still conveys the original point.

---

## Author Response (AR2)

Dear Dr Maussion -

Thank you for serving as the handling editor for our manuscript. We have uploaded our analysis scripts to Zenodo and added the following statement under *Code and data availability*: "The scripts and data used to generate all plots are archived at: https://doi.org/10.5281/zenodo.5707933."

Thanks, and kind regards -
Mark Flanner